

# Hybrid decision support system disaster management: application of lattice ordered q-rung linear Diophantine fuzzy hypersoft sets

J. Vimala[1], A. N. Surya[1], Nasreen Kausar[2], Dragan Pamucar[3,4,5], Seifedine Kadry[6,7] and Jungeun Kim[8]

[1] Department of Mathematics, Alagappa University, Karaikudi, Tamilnadu, India
[2] Department of Mathematics, Faculty of Arts and Science, Balikesir University, Balikesir, Turkey
[3] Department of Operations Research and Statistics, Faculty of Organizational Sciences, University of Belgrade, Belgrade, Serbia
[4] Department of Industrial Engineering and Management, Yuan Ze University, Taoyuan, Taiwan
[5] Department of Applied Mathematical Science, College of Science and Technology, Korea University, Sejong, Republic of South Korea
[6] Department of Computer Science and Mathematics, Lebanese American University, Beirut, Lebanon
[7] Noroff University College, Oslo, Norway
[8] Department of Computer Engineering, Inha University, Incheon, Republic of South Korea

Corresponding authors
J. Vimala,
vimaljey@alagappauniversity.ac.in
Jungeun Kim, jekim@inha.ac.kr

## ABSTRACT

The discovery of the lattice-ordered q-rung linear Diophantine fuzzy hypersoft set is a significant extension of fuzzy set theory. This study describes many of its fundamental algebraic operations, such as restricted union, extended union, restricted intersection, OR operation, and AND operation, along with examples. Further, an algorithm based on the proposed operations is presented in this study to handle multi-attributed decision-making problems extremely well, along with an illustrative multi-attribute decision-making example in the area of disaster management, which helps in choosing the most appropriate plan to tackle the known natural disaster by considering a greater number of attributes together. Further, the contribution of the method in the disaster management field is presented in the comparative analysis along with computational efficiency and scalability and an analysis of the comparison between the existing decision-making methods and the proposed one to express the superiority and advantages of the suggested approach over the existing methods.

## INTRODUCTION

The frequent occurrence of uncertainty-related issues in multi-attribute decision-making (MADM) makes them difficult to foresee and manage due to the extensive modeling of these uncertainties. The fuzzy set (FS) theory introduced by *Zadeh (1965)* is very useful for handling the difficulties brought on by uncertainty. However, FS only has a limited ability to reflect impartial situations. To overcome these restrictions, *Atanassov (1986)* devised the

notion of intuitionistic fuzzy sets (IFS). The IFS's two indices are membership degree (MD) and non-membership degree (NMD), and their sum value should fall within [0,1]. To solve problems smoothly, *Yager (2013)* developed the Pythagorean fuzzy set (PFS) in which the total of the $MD^2$ and $NMD^2$ should fall within [0,1]. *Yager (2016)* also proposed the q-rung orthopair fuzzy sets (q-ROFS), where the $MD^q$ and the $NMD^q$ are summed together and fall inside the range [0,1]. Later, various information measures (*Peng & Liu, 2019*) were proposed for q-ROFS. However, each of these ideas has drawbacks of its own. To overcome these drawbacks, *Riaz & Hashmi (2019)* formulated the theory of the linear Diophantine fuzzy set (LDFS), which contains the notion of reference parameters (RPs). Owing to the usefulness of LDFS, several researchers from various scientific fields were interested in them, and numerous significant studies were produced as a result (*Mahmood et al., 2021a*, *2021b*). Subsequently, the idea of quadratic diophantine fuzzy set was proposed by *Zia et al. (2023)*. Later, *Almagrabi et al. (2022)* created the q-rung linear Diophantine fuzzy set (q-RLDFS), a particular extension of the IFS, q-ROFS, and LDFS. Further, many real-world decision-making studies such as company selection problem (*Ali, 2025*), urban planning (*Petchimuthu et al., 2025*), logistics (*Kannan, Jayakumar & Pethaperumal, 2025*) and emerging technologies (*Kumar & Pamucar, 2025*). However, because they are not parametrized, each theory has drawbacks. To overcome the limitations brought on by parametrization, *Molodtsov (1999)* developed the idea of soft set (SS) theory, which handles vagueness in a parametric manner. Later, by incorporating FS and SS, *Roy & Maji (2007)* provided the idea of the fuzzy soft set (FSS), which helps present fuzzy data with parametric information. Similar to this, SS theory was incorporated with other extensions of FS theory such as IFS, PFS, q-ROFS, and LDFS (*Çağman & Karataş, 2013*; *Peng et al., 2015*; *Hussain et al., 2020*; *Riaz et al., 2020*) respectively, to exhibit these fuzzy extension data with parametric information and obtained intuitionistic fuzzy soft set (IFSS), Pythagorean fuzzy soft set (PFSS), q-rung orthopair fuzzy soft set (q-ROFSS) and linear Diophantine fuzzy soft set (LDFSS). *Smarandache (2018)* then transformed the function into a multi-attributed function to establish the idea of the hypersoft set (HSS) as an extension of SS. By incorporating HSS with FS and IFS, *Smarandache (2018)* also proposed the ideas of the fuzzy hypersoft set (FHSS) and intuitionistic fuzzy hypersoft set (IFHSS), which expresses FS and IFS data with multi-sub-parameter. Similarly, by incorporating q-ROFS with HSS, *Khan, Gulistan & Wahab (2022)* presented the q-rung orthopair fuzzy hypersoft set (q-ROFHSS), and by incorporating q-RLDFS with HSS, *Surya et al. (2024)* presented the q-rung linear Diophantine fuzzy hypersoft set (q-RLDFHSS). In many real-life problems, there is a ranking among the parameters to deal with such problems very effectively. *Ali et al. (2015)* proposed a lattice-ordered soft set (LOSS). Later, *Aslam et al. (2019)* discussed the notion of lattice-ordered fuzzy soft set (LOFSS), and *Mahmood et al. (2018)* discussed the notion of lattice-ordered intuitionistic fuzzy soft set (LOIFSS). Further, many researchers (*Rajareega & Vimala, 2021*; *Pandipriya, Vimala & Begam, 2018*; *Mahmood, Rehman & Sezgin, 2018*; *Begam et al., 2020*; *Khan, Bakhat & Iftikhar, 2019*; *Sabeena Begam & Vimala, 2019*) developed the concepts of lattice-ordered structure to various areas of FS theory and their extensions.

Likewise, to discuss real-life q-RLDFHS problems when there is a ranking among the multi-sub-parameters the notion of lattice ordered q-rung linear Diophantine fuzzy hypersoft set (LOq-RLDFHSS) is essential.

### Research gap

Listed below are the research gaps:

- From the analysis of existing literature, we can see that in theoretical aspects, the existing literature does not cover many fundamental algebraic operations of LOq-RLDFHSS.
- Further from the existing literature, we can see that while there are several parametric decision-making (DM) studies conducted under various fuzzy structures, it is challenging to demonstrate many MADM real-world problems under LOq-RLDFHS environment using the existing literature.

### Motivation

The following are the study's motivations:

- The study aims to close these research gaps by developing fundamental algebraic operations and a MADM method based on LOq-RLDFHSS.
- Another main motive of the study is to contribute to the disaster management field by the proposed MADM approach, since the existing DM methods in the disaster management field cannot handle multiple attributes simultaneously.

### Objectives

The main objectives of this work are listed below:

- To provide many fundamental algebraic operations of LOq-RLDFHSS.
- To provide an effective MADM strategy based on LOq-RLDFHSS.
- To provide an appropriate numerical illustration for the suggested MADM strategy in the field of disaster management.

### Contribution

The core contributions of the work are as follows:

- Many algebraic operators of LOq-RLDFHSS are proposed in this study, such as restricted union, restricted intersection, extended union, OR operation, AND operation, and complement.
- A MADM algorithm based on the LOq-RLDFHSS is presented in the study.
- Further, a real-world problem in the field of disaster management is depicted as a numerical example of the suggested MADM algorithm to show the efficacy of the proposed algorithm.
- To demonstrate the potency and efficacy of the suggested concepts and the MADM approach, a comparative assessment that describes the theoretical improvement of the proposed study and its contribution to the field of disaster management is presented, along with the minor restrictions of the proposed concepts.

The list of most of the abbreviations used in this study is given as a table in "List of abbreviation used in the study". The article is structured as follows:

"Background" contains the required introductory notations and definitions. "Algebraic operations of LOq-RLDFHSS" consists of fundamental algebraic operations of LOq-RLDFHSS. "MADM Approach Based on LOq-RLDFHSS" consists of a MADM algorithm based on LOq-RLDFHSS to successfully solve MADM challenges; a MADM problem in disaster management which demonstrates the efficiency of the proposed algorithm. To describe the superiority of the proposed idea to the existing ideas, a comparative assessment has been undertaken in "Comparative Assessment". Finally, "Conclusion" provides the conclusion of the article.

## BACKGROUND

This section provides the requisite notations and definitions for this article.

A binary relation $\leq$ on a non-empty set $\mathfrak{A}$ is said to be a partial order on $\mathfrak{A}$ if it is antisymmetric, reflexive and transitive. Also, $\leq$ is said to be a total order on $\mathfrak{A}$ if $\mathfrak{a} \neq \mathfrak{b}$, either $\mathfrak{a} \leq \mathfrak{b}$ or $\mathfrak{b} \leq \mathfrak{a} \ \forall \mathfrak{a}, \mathfrak{b} \in \mathfrak{A}$.

A partial order set L is said to be a lattice if the set $\{\mathfrak{a}, \mathfrak{b}\}$ has a greatest lower bound and least upper bound $\forall \mathfrak{a}, \mathfrak{b} \in$ L. If L contains 1 and 0 such that $\forall \mathfrak{x} \in$ L, $0 \leq \mathfrak{x} \leq 1$, then L is called a bounded lattice.

**Definition 2.1.** *Atanassov (1986)*: Let $\mathfrak{G}$ be the set of alternatives. A IFS I is defined as

$$I = \{(\mathfrak{g}, \Omega_I(\mathfrak{g}), \mho_I(\mathfrak{g})) | \mathfrak{g} \in \mathfrak{G}\}$$

where $\Omega_I(\mathfrak{g})$ and $\mho_I(\mathfrak{g}) \in [0,1]$ are MD and NMD fulfilling $0 \leq \Omega_I(\mathfrak{g}) + \mho_I(\mathfrak{g}) \leq 1$.

**Definition 2.2.** *Almagrabi et al. (2022)*: Let $\mathfrak{G}$ be the set of alternatives. A q-RLDFS Q is defined as

$$Q = \{(\mathfrak{g}, \langle \Omega_Q(\mathfrak{g}), \mho_Q(\mathfrak{g})\rangle, \langle \Delta_Q(\mathfrak{g}), \nabla_Q(\mathfrak{g})\rangle) | \mathfrak{g} \in \mathfrak{G}\}$$

where $\Omega_Q(\mathfrak{g}), \mho_Q(\mathfrak{g}), \Delta_Q(\mathfrak{g})$ and $\nabla_Q(\mathfrak{g}) \in [0,1]$ are MD, NMD and their corresponding RPs respectively, fulfilling $0 \leq \Delta_Q^q(\mathfrak{g}) + \nabla_Q^q(\mathfrak{g}) \leq 1$ and $0 \leq \Delta_Q^q(\mathfrak{g})\Omega_Q(\mathfrak{g}) + \nabla_Q^q(\mathfrak{g})\mho_Q(\mathfrak{g}) \leq 1 \ \forall \mathfrak{g} \in \mathfrak{G}$, q $\geq 1$.

**Definition 2.3.** *Molodtsov (1999)*: Let $\mathfrak{G}$ be the set of alternatives, $\mathscr{E}$ be the set of attributes, and $\mathscr{A} \subseteq \mathscr{E}$. Then SS is a pair $(\Theta, \mathscr{A})$ defined by the mapping

$$\Theta : \mathscr{A} \to P(\mathfrak{G})$$

where $P(\mathfrak{G})$ is the power set of $\mathfrak{G}$.

**Definition 2.4.** *Ali et al. (2015)*: Let $(\Theta, \mathscr{A})$ be a SS over $\mathfrak{G}$, where

$$\Theta : \mathscr{A} \to P(\mathfrak{G})$$

Then $(\Theta, \mathscr{A})$ is said to be a LOSS if $\mathfrak{a}_1 \leq_{\mathscr{A}} \mathfrak{a}_2 \Rightarrow \Theta(\mathfrak{a}_1) \subseteq \Theta(\mathfrak{a}_2) \forall \mathfrak{a}_1, \mathfrak{a}_2 \in \mathscr{A}$.

**Definition 2.5.** *Çağman & Karataş (2013)*: Let $\mathfrak{G}$ be the set of alternatives, $\mathscr{E}$ be the set of attributes, and $\mathscr{A} \subseteq \mathscr{E}$. Then IFSS is a pair $(\Theta, \mathscr{A})$ defined by the mapping

$$\Theta : \mathscr{A} \to IFP(\mathfrak{G})$$

where *IFP* $(\mathfrak{G})$ is the IF power set of $\mathfrak{G}$.

**Definition 2.6.** *Mahmood et al. (2018)*: Let $(\Theta, \mathscr{A})$ be a IFSS over $\mathfrak{G}$, where

$$\Theta : \mathscr{A} \to IFP(\mathfrak{G})$$

Then $(\Theta, \mathscr{A})$ is said to be a LOIFSS if $\mathfrak{a}_1 \leq_{\mathscr{A}} \mathfrak{a}_2 \Rightarrow \Theta(\mathfrak{a}_1) \subseteq \Theta(\mathfrak{a}_2) \forall \mathfrak{a}_1, \mathfrak{a}_2 \in \mathscr{A}$.

**Definition 2.7.** *Smarandache (2018)*: Let $\mathfrak{G}$ be the set of alternatives and $P(\mathfrak{G})$ denote the Power set of $\mathfrak{G}$. Let $\mathscr{E}_1, \mathscr{E}_2, ..., \mathscr{E}_n$ with $\mathscr{E}_i \cap \mathscr{E}_j = \varnothing$ for $i, j \in \{1, 2, ... n\}$ and $i \neq j$ be the attribute values of n distinct attributes $e_1, e_2, ..., e_n$ respectively and for each i = 1, 2, ... n, $\mathscr{A}_i$ be non empty subset of $\mathscr{E}_i$ and $\aleph_1 = \mathscr{A}_1 \times \mathscr{A}_2 \times ... \times \mathscr{A}_n \subseteq \mathscr{E}_1 \times \mathscr{E}_2 \times ... \times \mathscr{E}_n$. Then HSS over $\mathfrak{G}$ is the pair $(\Theta, \aleph_1)$ defined by the map

$$\Theta : \aleph_1 \to P(\mathfrak{G})$$

This can be represented as $(\Theta, \aleph_1) = \{(\eta, \Theta(\eta)) : \eta \in \aleph_1, \Theta(\eta) \in P(\mathfrak{G})\}$.

**Definition 2.8.** *Surya et al. (2024)*: Let $\mathfrak{G}$ be the set of alternatives and q-RLDFP $(\mathfrak{G})$ denote the q-RLDF Power set of $\mathfrak{G}$. Let $\mathscr{E}_1, \mathscr{E}_2, ..., \mathscr{E}_n$ with $\mathscr{E}_i \cap \mathscr{E}_j = \varnothing$ for $i, j \in \{1, 2, ... n\}$ and $i \neq j$ be the attribute values of n distinct attributes $e_1, e_2, ..., e_n$ respectively and for each i = 1, 2, ... n, $\mathscr{A}_i$ be non empty subset of $\mathscr{E}_i$ and $\aleph_1 = \mathscr{A}_1 \times \mathscr{A}_2 \times ... \times \mathscr{A}_n \subseteq \mathscr{E}_1 \times \mathscr{E}_2 \times ... \times \mathscr{E}_n$. Then, the q-Rung Linear Diophantine Fuzzy Hypersoft Set over $\mathfrak{G}$ (q-RLDFHSS $(\mathfrak{G})$) is the pair $(\Theta, \aleph_1)$ defined by the map

$$\Theta : \aleph_1 \to q - RLDFP(\mathfrak{G})$$

This can be represented as $(\Theta, \aleph_1) = \{(\eta, \Theta(\eta)) : \eta \in \aleph_1, \Theta(\eta) \in q - RLDFP(\mathfrak{G})\}$ and the q-RLDFHS Number (q-RLDFHSN) $\Theta_{\mathfrak{g}_a}(\eta_c) = \{\langle \Omega_{\Theta(\eta_c)}(\mathfrak{g}_a), \mho_{\Theta(\eta_c)}(\mathfrak{g}_a) \rangle, \langle \Delta_{\Theta(\eta_c)}(\mathfrak{g}_a), \nabla_{\Theta(\eta_c)}(\mathfrak{g}_a) \rangle | \mathfrak{g}_a \in \mathfrak{G} \ and \ \eta_c \in \aleph_1\}$ can be express as $\mathfrak{I}_{\eta_{ac}} = \{\langle \Omega_{\eta_{ac}}, \mho_{\eta_{ac}} \rangle, \langle \Delta_{\eta_{ac}}, \nabla_{\eta_{ac}} \rangle\}$.

**Definition 2.9.** *Surya et al. (2024)*: Let $(\Theta_1, \aleph_1), (\Theta_2, \aleph_2) \in$ q-RLDFHSS $(\mathfrak{G})$, then $(\Theta_1, \aleph_1)$ is said to be q-RLDFHS subset of $(\Theta_2, \aleph_2)$, if
   (i) $\aleph_1 \subseteq \aleph_2$
   (ii) $\forall \eta \in \aleph_1, \ \Theta_1(\eta) \subseteq \Theta_2(\eta)$
(i.e.,) $\Omega_{\Theta_1(\eta)}(\mathfrak{g}_a) \leq \Omega_{\Theta_2(\eta)}(\mathfrak{g}_a), \mho_{\Theta_2(\eta)}(\mathfrak{g}_a) \leq \mho_{\Theta_1(\eta)}(\mathfrak{g}_a), \Delta_{\Theta_1(\eta)}(\mathfrak{g}_a) \leq \Delta_{\Theta_2(\eta)}(\mathfrak{g}_a)$ and $\nabla_{\Theta_2(\eta)}(\mathfrak{g}_a) \leq \nabla_{\Theta_1(\eta)}(\mathfrak{g}_a) \forall \mathfrak{g}_a \in \mathfrak{G}$.

## ALGEBRAIC OPERATIONS OF LOQ-RLDFHSS

In this section, the fundamental algebraic operations of LOq-RLDFHSS are presented.

**Definition 3.1.** A *q-RLDFHSS* $(\mathfrak{G})$ $(\Theta, \mathscr{A}_1 \times \mathscr{A}_2 \times ... \times \mathscr{A}_n = \aleph_1)$ is said to be lattice ordered q-RLDFHSS over $\mathfrak{G}$ (LOq-RLDFHSS $(\mathfrak{G})$) if for mapping $\Theta : \aleph_1 \to q$-$RLDFP(\mathfrak{G})$,

$$\eta_1 \leq_{\aleph_1} \eta_2 \Rightarrow \Theta(\eta_1) \subseteq \Theta(\eta_2) \ \forall \ \eta_1, \eta_2 \in \aleph_1$$

$(i.e.,)$ $\eta_1 \leq_{\aleph_1} \eta_2$

$$\Rightarrow \Omega_{\Theta(\eta_1)}(\mathfrak{g}_a) \leq \Omega_{\Theta(\eta_2)}(\mathfrak{g}_a), \mho_{\Theta(\eta_2)}(\mathfrak{g}_a) \leq \mho_{\Theta(\eta_1)}(\mathfrak{g}_a),$$

$$\Delta_{\Theta(\eta_1)}(\mathfrak{g}_a) \leq \Delta_{\Theta(\eta_2)}(\mathfrak{g}_a) \text{ and } \nabla_{\Theta(\eta_2)}(\mathfrak{g}_a) \leq \nabla_{\Theta(\eta_1)}(\mathfrak{g}_a) \forall \mathfrak{g}_a \in \mathfrak{G}$$

where $\eta_1 = (\eta_{1_1}, \eta_{1_2}, \ldots, \eta_{1_n}), \eta_2 = (\eta_{2_1}, \eta_{2_2}, \ldots, \eta_{2_n})$ and $\eta_{1_i}, \eta_{2_i} \in \mathscr{A}_i$ for $i \in \{1, 2, \ldots, n\}$.

Also, each $\mathscr{A}_i$ is defined by its corresponding binary relation $\leq_{\mathscr{A}_i}$ and $\aleph_1$ forms a relation defined by $(\eta_{1_1}, \eta_{1_2}, \ldots, \eta_{1_n}) \leq_{\aleph_1} (\eta_{2_1}, \eta_{2_2}, \ldots, \eta_{2_n}) \Leftrightarrow \eta_{1_i} \leq_{\mathscr{A}_i} \eta_{2_i}$ in $\mathscr{A}_i$ for $i \in \{1, 2, \ldots, n\}$.

The following example clarifies the definition above.

**EXAMPLE 1.** Let $\mathfrak{G} = \{\mathfrak{g}_1, \mathfrak{g}_2, \mathfrak{g}_3\}$ be the set of hotels for accommodation, consider the attributes $e_1 = \{\text{charges}\}$, $e_2 = \{\text{food}\}$, $e_3 = \{\text{service}\}$ and $\mathscr{E}_1 = \{\text{extra charges } (e_{11}),$ room rent $(e_{12})\}$, $\mathscr{E}_2 = \{\text{taste } (e_{21}), \text{hygiene } (e_{22})\}$, $\mathscr{E}_3 = \{\text{customer service } (e_{31})\}$ be their corresponding attribute values respectively.

Suppose that,

For each i = 1, 2, 3, $\mathscr{A}_i = \mathscr{E}_i$

The elements in each set $\mathscr{A}_1, \mathscr{A}_2$ and $\mathscr{A}_3$ have an order among them, they are

The elements in set $\mathscr{A}_1$ are in the order $e_{11} \leq_{\mathscr{A}_1} e_{12}$

The elements in set $\mathscr{A}_2$ are in the order $e_{21} \leq_{\mathscr{A}_2} e_{22}$

$\mathscr{A}_3$ has only one element $e_{31}$ and

$$\aleph = \mathscr{A}_1 \times \mathscr{A}_2 \times \mathscr{A}_3 = \{\eta_1 = (e_{11}, e_{21}, e_{31}), \eta_2 = (e_{11}, e_{22}, e_{31}), \eta_3 = (e_{12}, e_{21}, e_{31}),$$
$$\eta_4 = (e_{12}, e_{22}, e_{31})\}$$

Then the order of elements in set $\aleph$ is shown in Fig. 1.

Further, the following is how the attributes are categorized

- The attribute "charges" and its attribute values indicates whether the alternative is cheap or not cheap
- The attribute "food" and its attribute values indicates whether the alternative is good or not good
- The attribute "service" and its attribute values indicates whether the alternative satisfies or dissatisfies

Then, the Cartesian product of attribute values exemplifies that the alternative is (cheap, good, satisfies) altogether or (not cheap, not good, dissatisfies) altogether.

Then, q-RLDFHSS $(\Theta, \aleph)$ may be expressed as

$$(\Theta, \aleph) = \left\{ \left\langle \eta_1, \left( \frac{\mathfrak{g}_1}{\langle (0.4, 0.8), (0.3, 0.9) \rangle}, \frac{\mathfrak{g}_2}{\langle (0.3, 0.7), (0.4, 0.9) \rangle}, \frac{\mathfrak{g}_3}{\langle (0.4, 0.7), (0.2, 0.7) \rangle} \right) \right\rangle, \right.$$

$$\left\langle \eta_2, \left( \frac{\mathfrak{g}_1}{\langle (0.4, 0.7), (0.4, 0.8) \rangle}, \frac{\mathfrak{g}_2}{\langle (0.4, 0.6), (0.5, 0.7) \rangle}, \frac{\mathfrak{g}_3}{\langle (0.5, 0.6), (0.4, 0.6) \rangle} \right) \right\rangle,$$

$$\left\langle \eta_3, \left( \frac{\mathfrak{g}_1}{\langle (0.5, 0.7), (0.5, 0.8) \rangle}, \frac{\mathfrak{g}_2}{\langle (0.4, 0.6), (0.5, 0.8) \rangle}, \frac{\mathfrak{g}_3}{\langle (0.4, 0.6), (0.3, 0.6) \rangle} \right) \right\rangle,$$

$$\left. \left\langle \eta_4, \left( \frac{\mathfrak{g}_1}{\langle (0.6, 0.6), (0.5, 0.6) \rangle}, \frac{\mathfrak{g}_2}{\langle (0.7, 0.6), (0.7, 0.5) \rangle}, \frac{\mathfrak{g}_3}{\langle (0.8, 0.4), (0.7, 0.3) \rangle} \right) \right\rangle \right\}$$

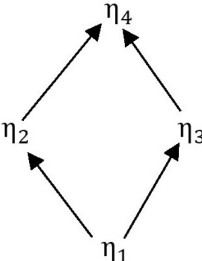

**Figure 1** **The order among elements in $\aleph$.**

We will assume that q = 3.

The characteristic of this q-RLDFHSS $(\Theta, \aleph)$ is $(\langle MD, NMD \rangle, \langle (\text{cheap, good, satisfies}), (\text{not cheap, not good, dissatisfies}) \rangle) \, \forall \eta_c \in \aleph$.

Clearly $\Theta(\eta_1) \subseteq \Theta(\eta_2) \subseteq \Theta(\eta_4)$ and $\Theta(\eta_1) \subseteq \Theta(\eta_3) \subseteq \Theta(\eta_4)$, therefore, $(\Theta, \aleph_1)$ is a LOq-RLDFHSS $(\mathfrak{G})$.

**Definition 3.2.** Let $\mathfrak{G}$ be the set of alternatives and $(\Theta_1, \aleph_1), (\Theta_2, \aleph_2) \in$ LOq-RLDFHSS $(\mathfrak{G})$. Their Restricted union is defined by $(\Theta_1, \aleph_1) \cup_{RES} (\Theta_2, \aleph_2) = (\Theta_3, \aleph_3)$ where $\aleph_3 = \aleph_1 \cap \aleph_2$ and $\forall \eta \in \aleph_3, \mathfrak{g} \in \mathfrak{G}$ we have $\Theta_1(\eta) \cup \Theta_2(\eta) = \Theta_3(\eta)$.

$$\Omega_{\Theta_3(\eta)}(\mathfrak{g}) = Max\{\Omega_{\Theta_1(\eta)}(\mathfrak{g}), \Omega_{\Theta_2(\eta)}(\mathfrak{g})\},$$
$$\mho_{\Theta_3(\eta)}(\mathfrak{g}) = Min\{\mho_{\Theta_1(\eta)}(\mathfrak{g}), \mho_{\Theta_2(\eta)}(\mathfrak{g})\},$$
$$\Delta_{\Theta_3(\eta)}(\mathfrak{g}) = Max\{\Delta_{\Theta_1(\eta)}(\mathfrak{g}), \Delta_{\Theta_2(\eta)}(\mathfrak{g})\} \text{ and}$$
$$\nabla_{\Theta_3(\eta)}(\mathfrak{g}) = Min\{\nabla_{\Theta_1(\eta)}(\mathfrak{g}), \nabla_{\Theta_2(\eta)}(\mathfrak{g})\}.$$

**Proposition 3.3.** *Let* $(\Theta_1, \aleph_1), (\Theta_2, \aleph_2) \in$ *LOq-RLDFHSS* $(\mathfrak{G})$. *Then* $(\Theta_1, \aleph_1) \cup_{RES} (\Theta_2, \aleph_2) \in$ *LOq-RLDFHSS* $(\mathfrak{G})$.

*Proof.* See "Proof of Proposition 3.2". □

**Definition 3.4.** Let $\mathfrak{G}$ be the set of alternatives and $(\Theta_1, \aleph_1), (\Theta_2, \aleph_2) \in$ LOq-RLDFHSS $(\mathfrak{G})$. Their Restricted intersection is defined by $(\Theta_1, \aleph_1) \cap_{RES} (\Theta_2, \aleph_2) = (\Theta_3, \aleph_3)$ where $\aleph_3 = \aleph_1 \cap \aleph_2$ and $\forall \eta \in \aleph_3, \mathfrak{g} \in \mathfrak{G}$ we have $\Theta_1(\eta) \cap \Theta_2(\eta) = \Theta_3(\eta)$.

$$\Omega_{\Theta_3(\eta)}(\mathfrak{g}) = Min\{\Omega_{\Theta_1(\eta)}(\mathfrak{g}), \Omega_{\Theta_2(\eta)}(\mathfrak{g})\},$$
$$\mho_{\Theta_3(\eta)}(\mathfrak{g}) = Max\{\mho_{\Theta_1(\eta)}(\mathfrak{g}), \mho_{\Theta_2(\eta)}(\mathfrak{g})\},$$
$$\Delta_{\Theta_3(\eta)}(\mathfrak{g}) = Min\{\Delta_{\Theta_1(\eta)}(\mathfrak{g}), \Delta_{\Theta_2(\eta)}(\mathfrak{g})\} \text{ and}$$
$$\nabla_{\Theta_3(\eta)}(\mathfrak{g}) = Max\{\nabla_{\Theta_1(\eta)}(\mathfrak{g}), \nabla_{\Theta_2(\eta)}(\mathfrak{g})\}.$$

**Proposition 3.5.** *Let* $(\Theta_1, \aleph_1), (\Theta_2, \aleph_2) \in$ *LOq-RLDFHSS* $(\mathfrak{G})$. *Then* $(\Theta_1, \aleph_1) \cap_{RES} (\Theta_2, \aleph_2) \in$ *LOq-RLDFHSS* $(\mathfrak{G})$.

*Proof.* See "Proof of Proposition 3.4". □

**Definition 3.6.** Let $\mathfrak{G}$ be the set of alternatives and $(\Theta_1, \aleph_1), (\Theta_2, \aleph_2) \in$ LOq-RLDFHSS $(\mathfrak{G})$. Their extended union is defined by $(\Theta_1, \aleph_1) \cup_{EXT} (\Theta_2, \aleph_2) = (\Theta_3, \aleph_3)$ where $\aleph_3 = \aleph_1 \cup \aleph_2$

$$(\Theta_3, \aleph_3) = \begin{cases} \{\langle\Omega_{\Theta_1(\eta)}(\mathfrak{g}), \mho_{\Theta_1(\eta)}(\mathfrak{g})\rangle, \langle\Delta_{\Theta_1(\eta)}(\mathfrak{g}), \nabla_{\Theta_1(\eta)}(\mathfrak{g})\rangle\} & \text{if } \eta \in \aleph_1 - \aleph_2 \\ \{\langle\Omega_{\Theta_2(\eta)}(\mathfrak{g}), \mho_{\Theta_2(\eta)}(\mathfrak{g})\rangle, \langle\Delta_{\Theta_2(\eta)}(\mathfrak{g}), \nabla_{\Theta_2(\eta)}(\mathfrak{g})\rangle\} & \text{if } \eta \in \aleph_2 - \aleph_1 \\ \{\langle Max\{\Omega_{\Theta_1(\eta)}(\mathfrak{g}), \Omega_{\Theta_2(\eta)}(\mathfrak{g})\}, Min\{\mho_{\Theta_1(\eta)}(\mathfrak{g}), \mho_{\Theta_2(\eta)}(\mathfrak{g})\}\rangle, & \text{if } \eta \in \aleph_1 \cap \aleph_2 \\ \langle Max\{\Delta_{\Theta_1(\eta)}(\mathfrak{g}), \Delta_{\Theta_2(\eta)}(\mathfrak{g})\}, Min\{\nabla_{\Theta_1(\eta)}(\mathfrak{g}), \nabla_{\Theta_2(\eta)}(\mathfrak{g})\}\rangle\} \end{cases}$$

**Proposition 3.7.** *Let* $(\Theta_1, \aleph_1), (\Theta_2, \aleph_2) \in$ *LOq-RLDFHSS* $(\mathfrak{G})$. *Then* $(\Theta_1, \aleph_1) \cup_{EXT}$ $(\Theta_2, \aleph_2) \in$ *LOq-RLDFHSS* $(\mathfrak{G})$, *if one of them is a LOq-RLDFHSS subset of other.*

*Proof.* See "Proof of Proposition 3.6". □

**Definition 3.8.** Let $\aleph_1, \aleph_2 \subseteq \mathscr{E}_1 \times \mathscr{E}_2 \times \ldots \times \mathscr{E}_n$. Then partial order $\leq_{\aleph_1 \times \aleph_2}$ on $\aleph_1 \times \aleph_2$ is defined as for any
$(\eta_1, \varsigma_1), (\eta_2, \varsigma_2) \in \aleph_1 \times \aleph_2, (\eta_1, \varsigma_1) \leq_{\aleph_1 \times \aleph_2} (\eta_2, \varsigma_2) \Leftrightarrow \eta_1 \leq_{\aleph_1} \eta_2$ and $\varsigma_1 \leq_{\aleph_2} \varsigma_2$.

**Definition 3.9.** Let $\mathfrak{G}$ be the set of alternatives and $(\Theta_1, \aleph_1), (\Theta_2, \aleph_2) \in$ LOq-RLDFHSS $(\mathfrak{G})$. Their "AND" operation is defined by $(\Theta_1, \aleph_1) \wedge (\Theta_2, \aleph_2) = (\Xi, \aleph_1 \times \aleph_2)$ where

$$\Xi(\aleph_1 \times \aleph_2) = \{(\eta, \varsigma), (\mathfrak{g}, \Xi(\eta, \varsigma)(\mathfrak{g})): \mathfrak{g} \in \mathfrak{G}, (\eta, \varsigma) \in \aleph_1 \times \aleph_2\}$$

and $\Xi(\eta, \varsigma)(\mathfrak{g}) = \{\langle Min\{\Omega_{\Theta_1(\eta)}(\mathfrak{g}), \Omega_{\Theta_2(\varsigma)}(\mathfrak{g})\}, Max\{\mho_{\Theta_1(\eta)}(\mathfrak{g}), \mho_{\Theta_2(\varsigma)}(\mathfrak{g})\}\rangle,$
$\langle Min\{\Delta_{\Theta_1(\eta)}(\mathfrak{g}), \Delta_{\Theta_2(\varsigma)}(\mathfrak{g})\}, Max\{\nabla_{\Theta_1(\eta)}(\mathfrak{g}), \nabla_{\Theta_2(\varsigma)}(\mathfrak{g})\}\rangle\}.$

**Proposition 3.10.** *Let* $\mathfrak{G}$ *be the set of alternatives and* $(\Theta_1, \aleph_1), (\Theta_2, \aleph_2) \in$ *LOq-RLDFHSS* $(\mathfrak{G})$. *Then* $(\Theta_1, \aleph_1) \wedge (\Theta_2, \aleph_2) \in$ *LOq-RLDFHSS* $(\mathfrak{G})$.

*Proof.* See "Proof of Proposition 3.9". □

**Definition 3.11.** Let $(\mathfrak{G})$ be the set of alternatives and $(\Theta_1, \aleph_1), (\Theta_2, \aleph_2) \in$ LOq-RLDFHSS $(\mathfrak{G})$. Then their "OR" operation is defined by $(\Theta_1, \aleph_1) \vee (\Theta_2, \aleph_2) = (\Xi, \aleph_1 \times \aleph_2)$ where

$(\Xi, \aleph_1 \times \aleph_2) = \{(\eta, \varsigma), (\mathfrak{g}, \Xi(\eta, \varsigma)(\mathfrak{g})): \mathfrak{g} \in \mathfrak{G}, (\eta, \varsigma) \in \aleph_1 \times \aleph_2\}$
and $\Xi(\eta, \varsigma)(\mathfrak{g}) = \{\langle Max\{\Omega_{\Theta_1(\eta)}(\mathfrak{g}), \Omega_{\Theta_2(\varsigma)}(\mathfrak{g})\}, Min\{\mho_{\Theta_1(\eta)}(\mathfrak{g}), \mho_{\Theta_2(\varsigma)}(\mathfrak{g})\}\rangle,$
$\langle Max\{\Delta_{\Theta_1(\eta)}(\mathfrak{g}), \Delta_{\Theta_2(\varsigma)}(\mathfrak{g})\}, Min\{\nabla_{\Theta_1(\eta)}(\mathfrak{g}), \nabla_{\Theta_2(\varsigma)}(\mathfrak{g})\}\rangle\}$

**Proposition 3.12.** *Let* $\mathfrak{G}$ *be the set of alternatives and* $(\Theta_1, \aleph_1), (\Theta_2, \aleph_2) \in$ *LOq-RLDFHSS* $(\mathfrak{G})$. *Then* $(\Theta_1, \aleph_1) \vee (\Theta_2, \aleph_2) \in$ *LOq-RLDFHSS* $(\mathfrak{G})$.

*Proof.* See "Proof of Proposition 3.11". □

**Definition 3.13.** Let $(\Theta_1, \aleph_1) \in$ LOq-RLDFHSS $(\mathfrak{G})$.
If $\Omega_{\Theta_1(\eta)}(\mathfrak{g}) = \Delta_{\Theta_1(\eta)}(\mathfrak{g}) = 0, \mho_{\Theta_1(\eta)}(\mathfrak{g}) = \nabla_{\Omega_1(\eta)}(\mathfrak{g}) = 1 \ \forall \eta \in \aleph_1$ and $\mathfrak{g} \in \mathfrak{G}$, Then, $(\Omega_1, \aleph_1)$ is called the relative null LOq-RLDFHSS and is denoted by $\varnothing_{\aleph_1}$.

**Definition 3.14.** Let $(\Theta_1, \aleph_1) \in$ LOq-RLDFHSS $(\mathfrak{G})$.

If $\Omega_{\Theta_1(\eta)}(\mathfrak{g}) = \Delta_{\Theta_1(\eta)}(\mathfrak{g}) = 1$, $\mho_{\Theta_1(\eta)}(\mathfrak{g}) = \nabla_{\Omega_1(\eta)}(\mathfrak{g}) = 0 \; \forall \eta \in \aleph_1$ and $\mathfrak{g} \in \mathfrak{G}$, Then, $(\Theta_1, \aleph_1)$ is called the relative universal LOq-RLDFHSS and is denoted by $\mathfrak{U}_{\aleph_1}$.

**Proposition 3.15.** *Let* $(\Theta_1, \aleph_1) \in$ *LOq-RLDFHSS* $(\mathfrak{G})$. *Then*

1. $(\Theta_1, \aleph_1) \cup_{RES} (\Theta_1, \aleph_1) = (\Theta_1, \aleph_1)$
2. $(\Theta_1, \aleph_1) \cup_{RES} \varnothing_{\aleph_1} = (\Theta_1, \aleph_1)$
3. $(\Theta_1, \aleph_1) \cup_{RES} \mathfrak{U}_{\aleph_1} = \mathfrak{U}_{\aleph_1}$
4. $(\Theta_1, \aleph_1) \cap_{RES} (\Theta_1, \aleph_1) = (\Theta_1, \aleph_1)$
5. $(\Theta_1, \aleph_1) \cap_{RES} \varnothing_{\aleph_1} = \varnothing_{\aleph_1}$
6. $(\Theta_1, \aleph_1) \cap_{RES} \mathfrak{U}_{\aleph_1} = (\Theta_1, \aleph_1)$

*Proof.* Straightforward. $\square$

**Definition 3.16.** Let $(\Theta_1, \aleph_1) \in$ LOq-RLDFHSS $(\mathfrak{G})$. Then complement of $(\Theta_1, \aleph_1)$ denoted by $(\Theta_1, \aleph_1)^c$ and is defined as follows
$(\Theta_1, \aleph_1)^c = \{(\mathfrak{g}, \{\langle \mho_{\Theta_1(\eta)}(\mathfrak{g}), \Omega_{\Theta_1(\eta)}(\mathfrak{g})\rangle, \langle \nabla_{\Theta_1(\eta)}(\mathfrak{g}), \Delta_{\Theta_1(\eta)}(\mathfrak{g})\rangle\}) : \eta \in \aleph_1$ and $\mathfrak{g} \in \mathfrak{G}\}$.

**Proposition 3.17.** *Let* $(\Theta_1, \aleph_1) \in$ *LOq-RLDFHSS* $(\mathfrak{G})$. *Then* $((\Theta_1, \aleph_1)^c)^c = (\Theta_1, \aleph_1)$

*Proof.* Let $(\Theta_1, \aleph_1) \in$ LOq-RLDFHSS $(\mathfrak{G})$. Then complement of $(\Theta_1, \aleph_1)$ is
$(\Theta_1, \aleph_1)^c = \{(\mathfrak{g}, \{\langle \mho_{\Theta_1(\eta)}(\mathfrak{g}), \Omega_{\Theta_1(\eta)}(\mathfrak{g})\rangle, \langle \nabla_{\Theta_1(\eta)}(\mathfrak{g}), \Delta_{\Theta_1(\eta)}(\mathfrak{g})\rangle\}) : \eta \in \aleph_1$ and $\mathfrak{g} \in \mathfrak{G}\}$,
  Now complement of $(\Theta_1, \aleph_1)^c$ is
$((\Theta_1, \aleph_1)^c)^c = \{(\mathfrak{g}, \{\langle \Omega_{\Theta_1(\eta)}(\mathfrak{g}), \mho_{\Theta_1(\eta)}(\mathfrak{g})\rangle, \langle \Delta_{\Theta_1(\eta)}(\mathfrak{g}), \nabla_{\Theta_1(\eta)}(\mathfrak{g})\rangle\}) : \eta \in \aleph_1$ and $\mathfrak{g} \in \mathfrak{G}\} = (\Theta_1, \aleph_1)$. $\square$

**EXAMPLE 2.** Let $\mathfrak{G} = \{\mathfrak{g}_1, \mathfrak{g}_2\}$ be a set of alternatives, $\aleph_1 = \{\eta_1, \eta_2\}$ be a set of parameters with an order defined by $\eta_1 \leq_\aleph \eta_2$ and $\aleph_2 = \{\eta_1, \eta_3\}$ be another set of parameters with an order defined by $\eta_1 \leq_\aleph \eta_3$. Then, let

$$(\Theta_1, \aleph_1) = \left\{ \left\langle \eta_1, \left( \frac{\mathfrak{g}_1}{\langle(0.3, 0.8), (0.4, 0.8)\rangle}, \frac{\mathfrak{g}_2}{\langle(0.2, 0.8), (0.2, 0.8)\rangle} \right) \right\rangle, \right.$$
$$\left. \left\langle \eta_2, \left( \frac{\mathfrak{g}_1}{\langle(0.6, 0.5), (0.5, 0.7)\rangle}, \frac{\mathfrak{g}_2}{\langle(0.6, 0.5), (0.6, 0.4)\rangle} \right) \right\rangle \right\}$$

be a q-RLDFHSS with q as 3, and since $\Theta_1(\eta_1) \subseteq \Theta_1(\eta_2)$, this implies $(\Theta_1, \aleph_1)$ is a LOq-RLDFHSS. Also, let

$$(\Theta_2, \aleph_2) = \left\{ \left\langle \eta_1, \left( \frac{\mathfrak{g}_1}{\langle(0.5, 0.8), (0.4, 0.7)\rangle}, \frac{\mathfrak{g}_2}{\langle(0.3, 0.7), (0.2, 0.8)\rangle} \right) \right\rangle, \right.$$
$$\left. \left\langle \eta_3, \left( \frac{\mathfrak{g}_1}{\langle(0.6, 0.7), (0.6, 0.6)\rangle}, \frac{\mathfrak{g}_2}{\langle(0.7, 0.3), (0.9, 0.2)\rangle} \right) \right\rangle \right\}$$

be another q-RLDFHSS with q as 3, and since $\Theta_2(\eta_1) \subseteq \Theta_2(\eta_3)$, this implies $(\Theta_2, \aleph_2)$ is a LOq-RLDFHSS.

The following operations are then derived:

- 

$$(\Theta_1, \aleph_1) \cup_{RES} (\Theta_2, \aleph_2) = \left\{ \left\langle \eta_1, \left( \frac{\mathfrak{g}_1}{\langle (0.5, 0.8), (0.4, 0.7) \rangle}, \frac{\mathfrak{g}_2}{\langle (0.3, 0.7), (0.2, 0.8) \rangle} \right) \right\rangle \right\}$$

- 

$$(\Theta_1, \aleph_1) \cap_{RES} (\Theta_2, \aleph_2) = \left\{ \left\langle \eta_1, \left( \frac{\mathfrak{g}_1}{\langle (0.3, 0.8), (0.4, 0.8) \rangle}, \frac{\mathfrak{g}_2}{\langle (0.2, 0.8), (0.2, 0.8) \rangle} \right) \right\rangle \right\}$$

- 

$$(\Theta_1, \aleph_1) \cup_{EXT} (\Theta_2, \aleph_2) = \left\{ \left\langle \eta_1, \left( \frac{\mathfrak{g}_1}{\langle (0.5, 0.8), (0.4, 0.7) \rangle}, \frac{\mathfrak{g}_2}{\langle (0.3, 0.7), (0.2, 0.8) \rangle} \right) \right\rangle, \right.$$
$$\left\langle \eta_2, \left( \frac{\mathfrak{g}_1}{\langle (0.6, 0.5), (0.5, 0.7) \rangle}, \frac{\mathfrak{g}_2}{\langle (0.6, 0.5), (0.6, 0.4) \rangle} \right) \right\rangle,$$
$$\left. \left\langle \eta_3, \left( \frac{\mathfrak{g}_1}{\langle (0.6, 0.7), (0.6, 0.6) \rangle}, \frac{\mathfrak{g}_2}{\langle (0.7, 0.3), (0.9, 0.2) \rangle} \right) \right\rangle \right\}$$

- 

$$(\Theta_1, \aleph_1) \vee (\Theta_2, \aleph_2) = \left\{ \left\langle (\eta_1, \eta_1), \left( \frac{\mathfrak{g}_1}{\langle (0.5, 0.8), (0.4, 0.7) \rangle}, \frac{\mathfrak{g}_2}{\langle (0.3, 0.7), (0.2, 0.8) \rangle} \right) \right\rangle, \right.$$
$$\left\langle (\eta_1, \eta_3), \left( \frac{\mathfrak{g}_1}{\langle (0.6, 0.7), (0.6, 0.7) \rangle}, \frac{\mathfrak{g}_2}{\langle (0.7, 0.3), (0.9, 0.2) \rangle} \right) \right\rangle,$$
$$\left\langle (\eta_2, \eta_1), \left( \frac{\mathfrak{g}_1}{\langle (0.6, 0.5), (0.5, 0.7) \rangle}, \frac{\mathfrak{g}_2}{\langle (0.6, 0.5), (0.6, 0.4) \rangle} \right) \right\rangle,$$
$$\left. \left\langle (\eta_2, \eta_3), \left( \frac{\mathfrak{g}_1}{\langle (0.6, 0.5), (0.6, 0.7) \rangle}, \frac{\mathfrak{g}_2}{\langle (0.7, 0.3), (0.9, 0.2) \rangle} \right) \right\rangle \right\}$$

- 

$$(\Theta_1, \aleph_1) \wedge (\Theta_2, \aleph_2) = \left\{ \left\langle (\eta_1, \eta_1), \left( \frac{\mathfrak{g}_1}{\langle (0.3, 0.8), (0.4, 0.8) \rangle}, \frac{\mathfrak{g}_2}{\langle (0.2, 0.8), (0.2, 0.8) \rangle} \right) \right\rangle, \right.$$
$$\left\langle (\eta_1, \eta_3), \left( \frac{\mathfrak{g}_1}{\langle (0.3, 0.8), (0.4, 0.8) \rangle}, \frac{\mathfrak{g}_2}{\langle (0.2, 0.8), (0.2, 0.8) \rangle} \right) \right\rangle,$$
$$\left\langle (\eta_2, \eta_1), \left( \frac{\mathfrak{g}_1}{\langle (0.5, 0.8), (0.4, 0.7) \rangle}, \frac{\mathfrak{g}_2}{\langle (0.3, 0.7), (0.2, 0.8) \rangle} \right) \right\rangle,$$
$$\left. \left\langle (\eta_2, \eta_3), \left( \frac{\mathfrak{g}_1}{\langle (0.6, 0.7), (0.5, 0.7) \rangle}, \frac{\mathfrak{g}_2}{\langle (0.6, 0.5), (0.6, 0.4) \rangle} \right) \right\rangle \right\}$$

## MADM APPROACH BASED ON LOQ-RLDFHSS

In this section, the comparison matrix of LOq-RLDFHSS and a MADM algorithm based on LOq-RLDFHSS are described and a MADM problem in the field of disaster management is discussed as a numerical illustration for the proposed MADM algorithm.

**Definition 4.1.** The comparison matrix of LOq-RLDFHSS is a matrix in which rows represent the alternatives such as $\mathfrak{g}_1, \mathfrak{g}_2, \ldots, \mathfrak{g}_m$ and columns represent the parameters

$\eta_1, \eta_2, \ldots, \eta_r$. The entries are $\mathfrak{h}_{ac}$ and computed as $\mathfrak{h}_{ac} = \frac{\mathfrak{k}_1 - \mathfrak{k}_2 + \mathfrak{k}_3 - \mathfrak{k}_4}{2}$, where $\mathfrak{k}_1$ is the integer computed as number of times $\Omega_{\Theta(\eta_c)}(\mathfrak{g}_a)$ greater than or equal to $\Omega_{\Theta(\eta_c)}(\mathfrak{g}_b)$, for $\mathfrak{g}_a \neq \mathfrak{g}_b, \forall \mathfrak{g}_b \in \mathfrak{G}$, $\mathfrak{k}_2$ is the integer computed as number of times $\mho_{\Theta(\eta_c)}(\mathfrak{g}_a)$ greater than or equal to $\mho_{\Theta(\eta_c)}(\mathfrak{g}_b)$, for $\mathfrak{g}_a \neq \mathfrak{g}_b, \forall \mathfrak{g}_b \in \mathfrak{G}$, $\mathfrak{k}_3$ is the integer computed as number of times $\Delta_{\Theta(\eta_c)}(\mathfrak{g}_a)$ greater than or equal to $\Delta_{\Theta(\eta_c)}(\mathfrak{g}_b)$, for $\mathfrak{g}_a \neq \mathfrak{g}_b, \forall \mathfrak{g}_b \in \mathfrak{G}$ and $\mathfrak{k}_4$ is the integer computed as number of times $\nabla_{\Theta(\eta_c)}(\mathfrak{g}_a)$ greater than or equal to $\nabla_{\Theta(\eta_c)}(\mathfrak{g}_b)$, for $\mathfrak{g}_a \neq \mathfrak{g}_b, \forall \mathfrak{g}_b \in \mathfrak{G}$. Further, the range of $\mathfrak{h}_{ac}$ lies within $[-(m-1), m-1]$.

**Definition 4.2.** The score of an alternative $\mathfrak{g}_a$ is $\mathfrak{S}_a$ and calculated as

$$\mathfrak{S}_a = \sum_{c=1}^{r} \mathfrak{h}_{ac}$$

where the range lies within $[r(-(m-1)), r(m-1)]$.

## Algorithm

The following steps describe the algorithm for selecting the most suitable alternative

Step 1: Consider the LOq-RLDFHSS ($\mathfrak{G}$) and keep it in tabular form

Step 2: Compute the comparison matrix of LOq-RLDFHSS.

Step 3: Calculate the score $\mathfrak{S}_a$ of $\mathfrak{g}_a \forall a$

Step 4: Find $\mathfrak{S}_l = \text{Max } \mathfrak{S}_a$ and choose it as the suitable alternative

Step 5: If multiple alternatives share the maximum score, select any one of them.

Figure 2 shows the proposed algorithm as a flowchart.

## Numerical illustration
### A general study about disaster management

Disaster management or emergency management is the administrative responsibility for creating the framework that assists societies in reducing their vulnerability to hazards and coping with calamities. Contrary to its name, disaster management does not focus on handling crises, which are typically regarded as minor occurrences with little consequences that are dealt with through regular community activities. The main goal of emergency management is the management of disasters, which are occurrences with more consequences than a community can manage on its own. A mix of efforts by individuals, households, businesses, local governments, and/or higher levels of government is typically required for disaster management. Even though the discipline of emergency management uses a variety of terminologies, operations can generally be broken down into four categories: preparedness, response, mitigation, and recovery. In other words, mitigation of disaster risks and prevention are also frequently used.

The guiding principle of disaster management is disaster mitigation. The continuous work aims to reduce disasters' harm to both persons and property. Mitigation measures include avoiding constructing near floodplains, designing bridges to resist earthquakes, developing and enforcing hurricane-proof building regulations, and more. Mitigation refers to sustained actions that minimize or prevent long-term danger to individuals and assets from environmental risks and their effects." Disaster consequences are continuously being lessened by federal, state, municipal, and individual actions.

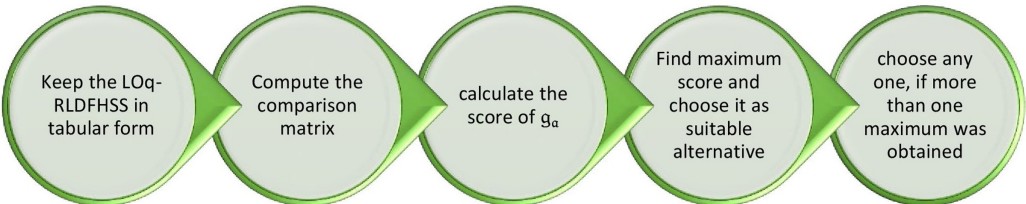

**Figure 2 Flowchart showing the steps of the proposed LOq-RLDFHSS-based MADM algorithm.**

Authorities and organizations on a national or international scale may provide this assistance during disaster. Effective coordination of disaster assistance is frequently crucial when numerous organizations contribute to the response, but competence has been degraded by the disaster or overwhelmed by demand. The US government released an article called the National Response Framework (*Federal Emergency Management Agency, 2023*) that outlines the responsibilities of authorities of the state, local, national, and tribal governments. It offers guidance on how to fully or partially implement disaster support services to aid in the response and recovery process.

The recovery phase begins once there is no immediate danger to human life. Getting the afflicted area back to normal as soon as possible is the urgent goal of the recovery phase. Trained laypeople give psychological first aid in the early wake of a disaster to help the affected populace cope and recover. In addition to providing practical support and assisting with procuring necessities like food and water, trained staff can also provide links to important resources. Similar to medical first aid, psychological first aid does not require therapists to be licensed clinicians. It is not debriefing, counseling, or psychotherapy.

Numerous research such as disaster management cycle of natural disaster (*Arifah, Tariq & Juni, 2019*), large scale group decision making in disaster management (*Wan et al., 2020*), post-disaster reconstruction projects (*Mohammadnazari et al., 2022*), use of indicators in vulnerability assessment (*Papathoma-Köhle et al., 2019*) in decision-making have been carried out in disaster management. Now, we show the utilization of proposed conceptions and algorithms in real life by a MADM problem in the field of disaster management, which helps to choose the most appropriate plan to tackle the known upcoming natural disaster by considering more attributes together. The problem is presented below, and its contribution to the disaster management field is discussed in detail in the comparative assessment section.

### Problem

Suppose disaster management wants to choose the most appropriate plan from a set of plans $\{\mathfrak{g}_1, \mathfrak{g}_2, \mathfrak{g}_3\}$ to tackle some of the known upcoming natural disasters as a precautionary measure and a team of decision makers was appointed to analyze the plans, the decision makers are considering the attributes $e_1 = \{\text{mitigation}\}$, $e_2 = \{\text{response}\}$, $e_3 = \{\text{recovery}\}$ and their sub attributes are $\mathscr{E}_1 = \{\text{education and awareness programs} (e_{11}), \text{regulation and infrastructure projects} (e_{12})\}$, $\mathscr{E}_2 = \{\text{maintaining regular services and activities} (e_{21}), \text{protecting life} (e_{22})\}$ and $\mathscr{E}_3 = \{\text{psychological recovery} (e_{31})\}$

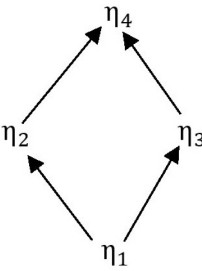

**Figure 3 The order among elements in $\aleph_1$.**

respectively. Also, the order of preference of elements in each set $\mathscr{E}_1$, $\mathscr{E}_2$ and $\mathscr{E}_3$ by decision-makers is given as follows

The elements in set $\mathscr{E}_1$ are in the order $e_{11} \leq_{\mathscr{E}_1} e_{12}$

The elements in set $\mathscr{E}_2$ are in the order $e_{21} \leq_{\mathscr{E}_2} e_{22}$

$\mathscr{E}_3$ has only one element $e_{31}$ and $\aleph_1 = \mathscr{E}_1 \times \mathscr{E}_2 \times \mathscr{E}_3 = \{\eta_1 = (e_{11}, e_{21}, e_{31}),$
$\eta_2 = (e_{11}, e_{22}, e_{31}), \eta_3 = (e_{12}, e_{21}, e_{31}), \eta_4 = (e_{12}, e_{22}, e_{31})\}$

Then the order of elements in set $\aleph_1$ is shown in the Fig. 3.

Further, decision-makers categorize the attributes as follows:

- The attribute "mitigation" and its attribute values indicates whether the plan is high or low

- The attribute "response" and its attribute values indicates whether the plan is good or not good

- The attribute "recovery" and its attribute values indicates whether the plan is effective or not effective

Then, the Cartesian product of attribute values exemplifies that the plan is (high, good, effective) all together or (low, not good, not effective) all together.

The opinions and data observed by the decision makers are constructed and expressed as a q-RLDFHSS $(\Theta, \aleph_1)$.

The characteristic of this q-RLDFHSS $(\Theta, \aleph_1)$ is $(\langle \text{MD, NMD} \rangle, \langle (\text{high, good, effective}),$ (low, not good, not effective)$\rangle) \; \forall \eta_c \in \aleph_1$.

$$(\Theta, \aleph_1) = \left\{ \left\langle \eta_1, \left( \frac{\mathfrak{g}_1}{\langle (0.33, 0.87), (0.31, 0.82) \rangle}, \frac{\mathfrak{g}_2}{\langle (0.29, 0.76), (0.33, 0.81) \rangle}, \frac{\mathfrak{g}_3}{\langle (0.38, 0.63), (0.17, 0.72) \rangle} \right) \right\rangle, \right.$$
$$\left\langle \eta_2, \left( \frac{\mathfrak{g}_1}{\langle (0.4, 0.65), (0.38, 0.71) \rangle}, \frac{\mathfrak{g}_2}{\langle (0.32, 0.57), (0.43, 0.66) \rangle}, \frac{\mathfrak{g}_3}{\langle (0.53, 0.61), (0.39, 0.51) \rangle} \right) \right\rangle,$$
$$\left\langle \eta_3, \left( \frac{\mathfrak{g}_1}{\langle (0.55, 0.66), (0.57, 0.72) \rangle}, \frac{\mathfrak{g}_2}{\langle (0.35, 0.53), (0.52, 0.81) \rangle}, \frac{\mathfrak{g}_3}{\langle (0.46, 0.54), (0.24, 0.57) \rangle} \right) \right\rangle,$$
$$\left. \left\langle \eta_4, \left( \frac{\mathfrak{g}_1}{\langle (0.63, 0.58), (0.65, 0.69) \rangle}, \frac{\mathfrak{g}_2}{\langle (0.63, 0.49), (0.74, 0.48) \rangle}, \frac{\mathfrak{g}_3}{\langle (0.83, 0.41), (0.72, 0.28) \rangle} \right) \right\rangle \right\}$$

We will assume that q = 3.

**Table 1 Tabular form of LOq-RLDFHSS $(\Theta, \aleph_1)$ which describes the data observed by the decision makers about the plans according to the parameters.**

| $(\Theta, \aleph_1)$ | $\mathfrak{g}_1$ | $\mathfrak{g}_2$ | $\mathfrak{g}_3$ |
|---|---|---|---|
| $\eta_1$ | $\langle(0.33, 0.87), (0.31, 0.82)\rangle$ | $\langle(0.29, 0.76), (0.33, 0.81)\rangle$ | $\langle(0.38, 0.63), (0.17, 0.72)\rangle$ |
| $\eta_2$ | $\langle(0.4, 0.65), (0.38, 0.71)\rangle$ | $\langle(0.32, 0.57), (0.43, 0.66)\rangle$ | $\langle(0.53, 0.61), (0.39, 0.51)\rangle$ |
| $\eta_3$ | $\langle(0.55, 0.66), (0.57, 0.72)\rangle$ | $\langle(0.35, 0.53), (0.52, 0.81)\rangle$ | $\langle(0.46, 0.54), (0.24, 0.57)\rangle$ |
| $\eta_4$ | $\langle(0.63, 0.58), (0.65, 0.69)\rangle$ | $\langle(0.63, 0.49), (0.74, 0.48)\rangle$ | $\langle(0.83, 0.41), (0.72, 0.28)\rangle$ |

**Table 2 Comparison matrix of LOq-RLDFHSS $(\Theta, \aleph_1)$.**

| $(\Theta, \aleph_1)$ | $\eta_1$ | $\eta_2$ | $\eta_3$ | $\eta_4$ |
|---|---|---|---|---|
| $\mathfrak{g}_1$ | $-1$ | $\frac{-3}{2}$ | $\frac{1}{2}$ | $\frac{-3}{2}$ |
| $\mathfrak{g}_2$ | $0$ | $\frac{1}{2}$ | $\frac{-1}{2}$ | $\frac{1}{2}$ |
| $\mathfrak{g}_3$ | $1$ | $1$ | $0$ | $\frac{3}{2}$ |

**Table 3 Score value of alternatives using the comparison matrix described in Table 2.**

| $(\Theta, \aleph_1)$ | $\mathfrak{g}_1$ | $\mathfrak{g}_2$ | $\mathfrak{g}_3$ |
|---|---|---|---|
| Score | $\frac{-7}{2}$ | $\frac{1}{2}$ | $\frac{7}{2}$ |

Clearly $\Theta(\eta_1) \subseteq \Theta(\eta_2) \subseteq \Theta(\eta_4)$ and $\Theta(\eta_1) \subseteq \Theta(\eta_3) \subseteq \Theta(\eta_4)$, therefore, $(\Theta, \aleph_1)$ is a LOq-RLDFHSS $(\mathfrak{G})$.

In this LOq-RLDFHSS, the plan $\mathfrak{g}_1$ and the parameter $\eta_1$ = (education and awareness programs, maintaining regular services and activities, psychological recovery) has the numeric value $\langle(0.33, 0.87), (0.31, 0.82)\rangle$. This value expresses that for the parameter $\eta_1$ the plan $\mathfrak{g}_1$ has 33% truth value and 87% false value. The pair (0.31,0.82) indicates the RP of the truth and false values, respectively, where we can observe that for (high at education and awareness programs, good at maintaining regular services and activities, effective in psychological recovery) all together the plan $\mathfrak{g}_1$ expresses 31% and for (low at education and awareness programs, not good at maintaining regular services and activities, not effective in psychological recovery) all together the plan $\mathfrak{g}_1$ expresses 82%. Similarly, all other numeric values are expressed in this LOq-RLDFHSS.

Step 1: Tabular form of LOq-RLDFHSS $(\Theta, \aleph_1)$ is shown in Table 1.

Step 2: Comparision matrix of LOq-RLDFHSS is shown in Table 2.

Step 3: The scores of the alternatives are shown in Table 3.

From the obtained scores, we observed that $\mathfrak{g}_3$ is the most appropriate plan to tackle the disaster, and we got the ranking of plans as $\mathfrak{g}_1 < \mathfrak{g}_2 < \mathfrak{g}_3$.

# COMPARATIVE ASSESSMENT

## Validity test

The effectiveness of a MADM strategy depends on the coherence of the qualities, the relationship between the alternatives, and the decision-maker's objective evaluations.

*Wang & Triantaphyllou (2008)* created three effective validity test criteria, which must be completed for a MADM approach to be considered legitimate.

**Test criteria 1:** The optimal choice remains the same if one selects a non-ideal alternative over a non-optimal one without changing the weight of any attribute.

**Test criteria 2:** The transitive nature is necessary for a decision-making approach to be effective.

**Test criteria 3:** If the decision-making problem is broken down into smaller subproblems, the smaller subproblem's order has to correspond with the original problem's order.

An examination of the suggested method's validity is provided below:

**Test criteria 1:** Consider the same disaster management problem by replacing the non-ideal alternative $\mathfrak{g}_1$ with a worse alternative $\widehat{\mathfrak{g}}$, whose numeric values according to the parameters are

$$\left\{ \left\langle \widehat{\mathfrak{g}}, \left( \frac{\eta_1}{\langle(0.30, 0.89), (0.25, 0.91)\rangle}, \frac{\eta_2}{\langle(0.35, 0.70), (0.33, 0.73)\rangle}, \right. \right. \right.$$
$$\left. \left. \left. \frac{\eta_3}{\langle(0.51, 0.68), (0.54, 0.76)\rangle}, \frac{\eta_4}{\langle(0.59, 0.62), (0.62, 0.72)\rangle} \right) \right\rangle \right\}.$$

Then, after analyzing these three alternatives $\widehat{\mathfrak{g}}, \ \mathfrak{g}_2, \ \mathfrak{g}_3$ by the proposed method, we obtain ranking as $\widehat{\mathfrak{g}} < \mathfrak{g}_2 < \mathfrak{g}_3$. The result makes it clear that the best solution remains constant. Therefore, test criteria 1 is satisfied for the proposed methodology.

**Test criteria 2 and 3:** We divide the considered problem into sub-problems as $\{\mathfrak{g}_1, \mathfrak{g}_3\}, \{\mathfrak{g}_1, \mathfrak{g}_2\}$ and $\{\mathfrak{g}_2, \mathfrak{g}_3\}$. Then using the proposed method we obtain $\mathfrak{g}_1 < \mathfrak{g}_3$, $\mathfrak{g}_1 < \mathfrak{g}_2$ and $\mathfrak{g}_2 < \mathfrak{g}_3$ as the ranking of sub-problems respectively. Therefore, we can see that the overall ranking remains constant as $\mathfrak{g}_1 < \mathfrak{g}_2 < \mathfrak{g}_3$. For the suggested approach, test criteria 2 and 3 are therefore valid.

### Comparative analysis

To analyze the superiority of the proposed DM method, the advantages and restrictions of existing and proposed DM methods are described in Table 4.

## DISCUSSION

### Superiority of the proposed MADM method

In Table 4, the comparison analysis brings to light the exceptional superiority of the innovatively proposed MADM method when juxtaposed with the array of existing MADM methodologies rooted in fuzzy theories such as FS, IFS, PFS, q-ROFS, LDFS, q-RLDFS, SS, FSS, IFSS, q-ROFSS, LDFSS, HSS, FHSS, IFHSS, q-ROFHSS, LOSS, LOFSS and LOIFSS. The distinguished LOq-RLDFHSS based DM method, in its unmatched prowess, showcases its ability to effectively manage q-RLDFS even within the intricate complexities of multi-sub-attributed scenarios that entail the prioritization and ordering of these multi-sub-attributes. This distinctive characteristic proves to be

**Table 4 Comparison table which describes the advantages and restrictions of existing and proposed decision making methods.**

| DM methods | Advantages | Restrictions |
|---|---|---|
| FS (*Zadeh, 1965*) | Addresses uncertainty by $\Omega$ (MD) | Unable to deal with $\mho$ (NMD) and parametrization |
| IFS (*Atanassov, 1986*) | Addresses uncertainty by $\Omega$ and $\mho$ | Restricted in handling uncertainty by the condition $\Omega + \mho \in [0,1]$, also unable to deal with parametrization |
| PFS (*Yager, 2013*) | Addresses uncertainty by $\Omega$ and $\mho$ even if $\Omega + \mho \notin [0,1]$ | Restricted in handling uncertainty by the condition $\Omega^2 + \mho^2 \in [0,1]$, also unable to deal with parametrization |
| q-ROFS (*Yager, 2016*) | Addresses uncertainty by $\Omega$ and $\mho$ even if $\Omega^2 + \mho^2 \notin [0,1]$ | Restricted in handling uncertainty by the condition $\Omega^q + \mho^q \in [0,1]$, also unable to deal with parametrization |
| LDFS (*Riaz & Hashmi, 2019*) | Addresses uncertainty by $\Omega$, $\mho$, $\Delta$ (RP corresponding to MD) and $\nabla$ (RP corresponding to NMD) even if $\Omega^q + \mho^q \notin [0,1]$ | Restricted in handling uncertainty by the conditions $\Delta\Omega + \nabla\mho \in [0,1]$ and $\Delta + \nabla \in [0,1]$, also unable to deal with parametrization |
| q-RLDFS (*Almagrabi et al., 2022*) | Addresses uncertainty by $\Omega$, $\mho$, $\Delta$ and $\nabla$ even if $\Delta\Omega + \nabla\mho \notin [0,1]$ and $\Delta + \nabla \notin [0,1]$ | Restricted in handling uncertainty by the conditions $\Delta^q\Omega + \nabla^q\mho \in [0,1]$ and $\Delta^q + \nabla^q \in [0,1]$ also unable to deal with parametrization |
| SS (*Molodtsov, 1999*) | Able to deal with parametrization | Unable to address uncertainty by parameterization |
| FSS (*Roy & Maji, 2007*) | Addresses FS with parameterized values | Unable to address uncertainty exceeding FS's restriction by parameterized values and also unable to address FS by multi-sub-parameterized values |
| IFSS (*Çağman & Karataş, 2013*) | Addresses IFS with parameterized values | Unable to address uncertainty exceeding IFS's restriction by parameterized values and also unable to address IFS by multi-sub-parameterized values |
| q-ROFSS (*Hussain et al., 2020*) | Addresses q-ROFS with parameterized values | Unable to address uncertainty exceeding q-ROFS's restriction by parameterized values and also unable to address q-ROFS by multi-sub-parameterized values |
| LDFSS (*Riaz et al., 2020*) | Addresses LDFS with parameterized values | Unable to address uncertainty exceeding LDFS's restriction by parameterized values and also unable to address LDFS by multi-sub-parameterized values |
| LOSS (*Ali et al., 2015*) | Addresses SS effectively when there is a ranking among parameters | Unable to address uncertainty by parameterization |
| LOFSS (*Aslam et al., 2019*) | Addresses FSS effectively when there is a ranking among parameters | Unable to address uncertainty exceeding FS's restriction by parameterized values and also unable to address FS by multi-sub-parameterized values |
| LOIFSS (*Mahmood et al., 2018*) | Addresses IFSS effectively when there is a ranking among parameters | Unable to address uncertainty exceeding IFS's restriction by parameterized values and also unable to address IFS by multi-sub-parameterized values |
| HSS (*Smarandache, 2018*) | Able to deal with multi-sub-parametrization | Unable to address uncertainty by multi-sub-parameterization |
| FHSS (*Smarandache, 2018*) | Addresses FS with multi-sub-parameterized values | Unable to address uncertainty exceeding FS's restriction by multi-sub-parameterized values |
| IFHSS (*Smarandache, 2018*) | Addresses IFS with multi-sub-parameterized values | Unable to address uncertainty exceeding IFS's restriction by multi-sub-parameterized values |
| q-ROFHSS (*Khan, Gulistan & Wahab, 2022*) | Addresses q-ROFS with multi-sub-parameterized values | Unable to address uncertainty exceeding q-ROFS's restriction by multi-sub-parameterized values |
| q-RLDFHSS (*Surya et al., 2024*) | Addresses q-RLDFS with multi-sub-parameterized values | Unable to address uncertainty exceeding q-RLDFS's restriction by multi-sub-parameterized values |
| LOq-RLDFHSS (proposed) | Addresses q-RLDFHSS effectively even when there is a ranking among multi-sub-parameters | Unable to address uncertainty exceeding q-RLDFS's restriction by multi-sub-parameterized values |

more suitable in navigating through a wide spectrum of real-world MADM situations with finesse.

## Computational effiency and scalability

The proposed MADM is highly efficient in terms of scalability since, it is capable of handling real-world problems with large data. Further, the proposed MADM methodology is capable of handling problems with large number of alternatives and parameters, but to understand the methodology clearly, the disaster management problem given in Section "MADM Approach Based on LOq-RLDFHSS" considers three alternatives and four parameters. Also, it is suitable to implement the proposed MADM method in various large-scale real-world applications such as medical diagnosis, supply chain optimization and more. In this study it is contributed to the field of disaster management. Also, the results obtained by the proposed method is more reliable and accurate since it considers more parameters and data, to handle the problem than the existing fuzzy MADM methods.

## Contribution in the disaster management field

Even though various decision-making approaches and case studies (*Arifah, Tariq & Juni, 2019*; *Wan et al., 2020*; *Mohammadnazari et al., 2022*; *Papathoma-Köhle et al., 2019*) contribute to disaster management, those studies became inadequate when the disaster situation needed to incorporate more attributes together simultaneously to obtain the most appropriate solution. Further, the presented case study is a unique case in disaster management, which is not yet and unable to be described by the existing MADM approaches in the disaster management field. From this it becomes clear that conventional MADM strategies are inadequate when confronted with scenarios teeming with many intricate data, unlike our proposed method, which adeptly converts intricate parameter data into streamlined numerical formats.

Also, it is crucial to recognize that while the proposed method undeniably offers substantial benefits, it is not devoid of its own set of limitations, such as limitations mentioned in Table 4. Further, the algorithm shows a limitation in the case of ties.

## CONCLUSION

For addressing a wide range of uncertain challenges, the q-RLDFHSS and LOq-RLDFHSS stand out as innovative extensions of FS theory. Throughout this research, numerous fundamental algebraic operations of LOq-RLDFHSS have been identified, emphasizing the development of an algorithm specifically designed to solve MADM problems leveraging the concepts of LOq-RLDFHSS. By exploring a unique MADM scenario within the domain of disaster management, which helps to choose the most appropriate plan to tackle the known upcoming natural disaster by considering more attributes together, the use of the suggested method in practice is thoroughly examined. The comparative analysis showcases the superiority and effectiveness of the novel MADM method against existing approaches, underscoring its value in real-world applications. In the comparative analysis, the study's contribution to the disaster management field is also discussed in detail.

**Table 5 List of abbreviation used in the study.**

| Abbreviation | Description |
|---|---|
| FS | Fuzzy set |
| MADM | Multi-attributed decision-making |
| MD | Membership degree |
| IFS | Intuitionistic fuzzy set |
| NMG | Non-membership Degree |
| PFS | Pythagorean fuzzy set |
| q-ROFS | q-Rung orthopair fuzzy set |
| LDFS | Linear Diophantine fuzzy set |
| RPs | Reference parameters |
| q-RLDFS | q-Rung linear Diophantine fuzzy set |
| SS | Soft set |
| FSS | Fuzzy soft set |
| IFSS | Intuitonistic fuzzy soft set |
| q-ROFSS | q-Rung orthopair fuzzy soft set |
| LDFSS | Linear Diophantine fuzzy soft set |
| HSS | Hypersoft set |
| FHSS | Fuzzy hypersoft set |
| IFHSS | Intuitionistic fuzzy hypersoft set |
| q-ROFHSS | q-Rung orthopair fuzzy hypersoft set |
| q-RLDFHSS | q-Rung linear diophantine fuzzy hypersoft set |
| LOSS | Lattice ordered soft set |
| LOFSS | Lattice ordered fuzzy soft set |
| LOIFSS | Lattice ordered intuitionistic fuzzy soft set |
| LOq-RLDFSS | Lattice ordered q-rung linear Diophantine fuzzy hypersoft set |

## Future direction

In future, it will focus on developing advanced information measures and aggregation operators tailored for the LOq-RLDFHSS. Further, it will be focused on overcoming the limitations of the proposed study by utilizing the concept of hesitancy function described in *Zia et al. (2024)*. Also, it is aimed to discuss various real-world problems in different domains such as medical, cybersecurity and pattern recognition.

## APPENDIX

### List of abbreviation used in the study

The list of most of the abbreviations used in this study is described in Table 5.

### Proof of proposition 3.2

*Proof.* Let $(\Theta_1, \aleph_1), (\Theta_2, \aleph_2) \in$ LOq-RLDFHSS $(\mathfrak{G})$. Then by Definition 3.2
$\Theta_1(\eta) \cup \Theta_2(\eta) = \Theta_3(\eta)$, where $\eta \in \aleph_3 = \aleph_1 \cap \aleph_2$.
If $\aleph_1 \cap \aleph_2 = \varnothing$, then result is trivial.
Now for $\aleph_1 \cap \aleph_2 \neq \varnothing$, since $\aleph_1, \aleph_2 \subseteq \mathscr{E}_1 \times \mathscr{E}_2 \times \ldots \times \mathscr{E}_n$

Therefore for any $\eta_c \leq_{\aleph_1} \eta_d$ we have $\Theta_1(\eta_c) \subseteq \Theta_1(\eta_d), \forall \eta_c, \eta_d \in \aleph_1$
and for any $\varsigma_c \leq_{\aleph_2} \varsigma_d$ we have $\Theta_2(\varsigma_c) \subseteq \Theta_2(\varsigma_d), \forall \varsigma_c, \varsigma_d \in \aleph_2$
Now for any $\varpi_c, \varpi_d \in \aleph_3$ and $\varpi_c \leq_{\aleph_3} \varpi_d$

$\Rightarrow \varpi_c, \varpi_d \in \aleph_1 \cap \aleph_2$

$\Rightarrow \varpi_c, \varpi_d \in \aleph_1$ and $\varpi_c, \varpi_d \in \aleph_2$

$\Rightarrow \Theta_1(\varpi_c) \subseteq \Theta_1(\varpi_d)$ and $\Theta_2(\varpi_c) \subseteq \Theta_2(\varpi_d)$ whenever $\varpi_c \leq_{\aleph_1} \varpi_d, \varpi_c \leq_{\aleph_2} \varpi_d$

$\Rightarrow \Omega_{\Theta_1(\varpi_c)}(\mathfrak{g}) \leq \Omega_{\Theta_1(\varpi_d)}(\mathfrak{g}), \Omega_{\Theta_2(\varpi_c)}(\mathfrak{g}) \leq \Omega_{\Theta_2(\varpi_d)}(\mathfrak{g})$
$\quad \mho_{\Theta_1(\varpi_d)}(\mathfrak{g}) \leq \mho_{\Theta_1(\varpi_c)}(\mathfrak{g}), \mho_{\Theta_2(\varpi_d)}(\mathfrak{g}) \leq \mho_{\Theta_2(\varpi_c)}(\mathfrak{g})$
$\quad \Delta_{\Theta_1(\varpi_c)}(\mathfrak{g}) \leq \Delta_{\Theta_1(\varpi_d)}(\mathfrak{g}), \Delta_{\Theta_2(\varpi_c)}(\mathfrak{g}) \leq \Delta_{\Theta_2(\varpi_d)}(\mathfrak{g})$
$\quad \nabla_{\Theta_1(\varpi_d)}(\mathfrak{g}) \leq \nabla_{\Theta_1(\varpi_c)}(\mathfrak{g}), \nabla_{\Theta_2(\varpi_d)}(\mathfrak{g}) \leq \nabla_{\Theta_2(\varpi_c)}(\mathfrak{g})$

$\Rightarrow \text{Max}\{\Omega_{\Theta_1(\varpi_c)}(\mathfrak{g}), \Omega_{\Theta_2(\varpi_c)}(\mathfrak{g})\} \leq \text{Max}\{\Omega_{\Theta_1(\varpi_d)}(\mathfrak{g}), \Omega_{\Theta_2(\varpi_d)}(\mathfrak{g})\}$
$\quad \text{Min}\{\mho_{\Theta_1(\varpi_d)}(\mathfrak{g}), \mho_{\Theta_2(\varpi_d)}(\mathfrak{g})\} \leq \text{Min}\{\mho_{\Theta_1(\varpi_c)}(\mathfrak{g}), \mho_{\Theta_2(\varpi_c)}(\mathfrak{g})\}$
$\quad \text{Max}\{\Delta_{\Theta_1(\varpi_c)}(\mathfrak{g}), \Delta_{\Theta_2(\varpi_c)}(\mathfrak{g})\} \leq \text{Max}\{\Delta_{\Theta_1(\varpi_d)}(\mathfrak{g}), \Delta_{\Theta_2(\varpi_d)}(\mathfrak{g})\}$
$\quad \text{Min}\{\nabla_{\Theta_1(\varpi_d)}(\mathfrak{g}), \nabla_{\Theta_2(\varpi_d)}(\mathfrak{g})\} \leq \text{Min}\{\nabla_{\Theta_1(\varpi_c)}(\mathfrak{g}), \nabla_{\Theta_2(\varpi_c)}(\mathfrak{g})\}$

$\Rightarrow \Omega_{\Theta_1(\varpi_c) \cup \Theta_2(\varpi_c)}(\mathfrak{g}) \leq \Omega_{\Theta_1(\varpi_d) \cup \Theta_2(\varpi_d)}(\mathfrak{g})$
$\quad \mho_{\Theta_1(\varpi_d) \cup \Theta_2(\varpi_d)}(\mathfrak{g}) \leq \mho_{\Theta_1(\varpi_c) \cup \Theta_2(\varpi_c)}(\mathfrak{g})$
$\quad \Delta_{\Theta_1(\varpi_c) \cup \Theta_2(\varpi_c)}(\mathfrak{g}) \leq \Delta_{\Theta_1(\varpi_d) \cup \Theta_2(\varpi_d)}(\mathfrak{g})$
$\quad \nabla_{\Theta_1(\varpi_d) \cup \Theta_2(\varpi_d)}(\mathfrak{g}) \leq \nabla_{\Theta_1(\varpi_c) \cup \Theta_2(\varpi_c)}(\mathfrak{g})$

$\Rightarrow \Omega_{\Theta_3(\varpi_c)}(\mathfrak{g}) \leq \Omega_{\Theta_3(\varpi_d)}(\mathfrak{g})$
$\quad \mho_{\Theta_3(\varpi_d)}(\mathfrak{g}) \leq \mho_{\Theta_3(\varpi_c)}(\mathfrak{g})$
$\quad \Delta_{\Theta_3(\varpi_c)}(\mathfrak{g}) \leq \Delta_{\Theta_3(\varpi_d)}(\mathfrak{g})$
$\quad \nabla_{\Theta_3(\varpi_d)}(\mathfrak{g}) \leq \nabla_{\Theta_3(\varpi_c)}(\mathfrak{g})$

$\Rightarrow \Theta_3(\varpi_c) \subseteq \Theta_3(\varpi_d)$ for $\varpi_c \leq_{\aleph_3} \varpi_d$

$\Rightarrow (\Theta_1, \aleph_1) \cup_{RES} (\Theta_2, \aleph_2) \in$ LOq-RLDFHSS $(\mathfrak{G})$. $\square$

## Proof of proposition 3.4

*Proof.* Let $(\Theta_1, \aleph_1), (\Theta_2, \aleph_2) \in$ LOq-RLDFHSS $(\mathfrak{G})$. Then by Definition 3.4
$\quad \Theta_1(\eta) \cap \Theta_2(\eta) = \Theta_3(\eta)$, where $\eta \in \aleph_3 = \aleph_1 \cap \aleph_2$.
If $\aleph_1 \cap \aleph_2 = \varnothing$, then result is trivial.
Now for $\aleph_1 \cap \aleph_2 \neq \varnothing$, since $\aleph_1, \aleph_2 \subseteq \mathscr{E}_1 \times \mathscr{E}_2 \times \ldots \times \mathscr{E}_n$
Therefore for any $\eta_c \leq_{\aleph_1} \eta_d$ we have $\Theta_1(\eta_c) \subseteq \Theta_1(\eta_d), \forall \eta_c, \eta_d \in \aleph_1$
and for any $\varsigma_c \leq_{\aleph_2} \varsigma_d$ we have $\Theta_2(\varsigma_c) \subseteq \Theta_2(\varsigma_d), \forall \varsigma_c, \varsigma_d \in \aleph_2$
Now for any $\varpi_c, \varpi_d \in \aleph_3$ and $\varpi_c \leq_{\aleph_3} \varpi_d$

$\Rightarrow \varpi_c, \varpi_d \in \aleph_1 \cap \aleph_2$

$\Rightarrow \varpi_c, \varpi_d \in \aleph_1$ and $\varpi_c, \varpi_d \in \aleph_2$

$\Rightarrow \Theta_1(\varpi_c) \subseteq \Theta_1(\varpi_d)$ and $\Theta_2(\varpi_c) \subseteq \Theta_2(\varpi_d)$ whenever $\varpi_c \leq_{\aleph_1} \varpi_d, \varpi_c \leq_{\aleph_2} \varpi_d$

$\Rightarrow \Omega_{\Theta_1(\varpi_c)}(\mathfrak{g}) \leq \Omega_{\Theta_1(\varpi_d)}(\mathfrak{g}), \Omega_{\Theta_2(\varpi_c)}(\mathfrak{g}) \leq \Omega_{\Theta_2(\varpi_d)}(\mathfrak{g})$
$\quad \mho_{\Theta_1(\varpi_d)}(\mathfrak{g}) \leq \mho_{\Theta_1(\varpi_c)}(\mathfrak{g}), \mho_{\Theta_2(\varpi_d)}(\mathfrak{g}) \leq \mho_{\Theta_2(\varpi_c)}(\mathfrak{g})$
$\quad \Delta_{\Theta_1(\varpi_c)}(\mathfrak{g}) \leq \Delta_{\Theta_1(\varpi_d)}(\mathfrak{g}), \Delta_{\Theta_2(\varpi_c)}(\mathfrak{g}) \leq \Delta_{\Theta_2(\varpi_d)}(\mathfrak{g})$
$\quad \nabla_{\Theta_1(\varpi_d)}(\mathfrak{g}) \leq \nabla_{\Theta_1(\varpi_c)}(\mathfrak{g}), \nabla_{\Theta_2(\varpi_d)}(\mathfrak{g}) \leq \nabla_{\Theta_2(\varpi_c)}(\mathfrak{g})$

$$\Rightarrow \mathrm{Min}\{\Omega_{\Theta_1(\varpi_c)}(\mathfrak{g}), \Omega_{\Theta_2(\varpi_c)}(\mathfrak{g})\} \leq \mathrm{Min}\{\Omega_{\Theta_1(\varpi_d)}(\mathfrak{g}), \Omega_{\Theta_2(\varpi_d)}(\mathfrak{g})\}$$

$$\mathrm{Max}\{\mho_{\Theta_1(\varpi_d)}(\mathfrak{g}), \mho_{\Theta_2(\varpi_d)}(\mathfrak{g})\} \leq \mathrm{Max}\{\mho_{\Theta_1(\varpi_c)}(\mathfrak{g}), \mho_{\Theta_2(\varpi_c)}(\mathfrak{g})\}$$

$$\mathrm{Min}\{\Delta_{\Theta_1(\varpi_c)}(\mathfrak{g}), \Delta_{\Theta_2(\varpi_c)}(\mathfrak{g})\} \leq \mathrm{Min}\{\Delta_{\Theta_1(\varpi_d)}(\mathfrak{g}), \Delta_{\Theta_2(\varpi_d)}(\mathfrak{g})\}$$

$$\mathrm{Max}\{\nabla_{\Theta_1(\varpi_d)}(\mathfrak{g}), \nabla_{\Theta_2(\varpi_d)}(\mathfrak{g})\} \leq \mathrm{Max}\{\nabla_{\Theta_1(\varpi_c)}(\mathfrak{g}), \nabla_{\Theta_2(\varpi_c)}(\mathfrak{g})\}$$

$$\Rightarrow \Omega_{\Theta_1(\varpi_c) \cap \Theta_2(\varpi_c)}(\mathfrak{g}) \leq \Omega_{\Theta_1(\varpi_d) \cap \Theta_2(\varpi_d)}(\mathfrak{g})$$

$$\mho_{\Theta_1(\varpi_d) \cap \Theta_2(\varpi_d)}(\mathfrak{g}) \leq \mho_{\Theta_1(\varpi_c) \cap \Theta_2(\varpi_c)}(\mathfrak{g})$$

$$\Delta_{\Theta_1(\varpi_c) \cap \Theta_2(\varpi_c)}(\mathfrak{g}) \leq \Delta_{\Theta_1(\varpi_d) \cap \Theta_2(\varpi_d)}(\mathfrak{g})$$

$$\nabla_{\Theta_1(\varpi_d) \cap \Theta_2(\varpi_d)}(\mathfrak{g}) \leq \nabla_{\Theta_1(\varpi_c) \cap \Theta_2(\varpi_c)}(\mathfrak{g})$$

$$\Rightarrow \Omega_{\Theta_3(\varpi_c)}(\mathfrak{g}) \leq \Omega_{\Theta_3(\varpi_d)}(\mathfrak{g})$$

$$\mho_{\Theta_3(\varpi_d)}(\mathfrak{g}) \leq \mho_{\Theta_3(\varpi_c)}(\mathfrak{g})$$

$$\Delta_{\Theta_3(\varpi_c)}(\mathfrak{g}) \leq \Delta_{\Theta_3(\varpi_d)}(\mathfrak{g})$$

$$\nabla_{\Theta_3(\varpi_d)}(\mathfrak{g}) \leq \nabla_{\Theta_3(\varpi_c)}(\mathfrak{g})$$

$$\Rightarrow \Theta_3(\varpi_c) \subseteq \Theta_3(\varpi_d) \text{ for } \varpi_c \leq_{\aleph_3} \varpi_d$$

$$\Rightarrow (\Theta_1, \aleph_1) \cap_{RES} (\Theta_2, \aleph_2) \in \text{LOq-RLDFHSS } (\mathfrak{G}). \ \square$$

**Proof of proposition 3.6**

*Proof.* Let $(\Theta_1, \aleph_1), (\Theta_2, \aleph_2) \in \text{LOq-RLDFHSS}(\mathfrak{G})$. Then by Definition 3.6

$$(\Theta_1, \aleph_1) \cup_{EXT} (\Theta_2, \aleph_2) = (\Theta_3, \aleph_3) \text{ where } \aleph_3 = \aleph_1 \cup \aleph_2$$

$$(\Theta_3, \aleph_3) = \begin{cases} \{\langle \Omega_{\Theta_1(\eta)}(\mathfrak{g}), \mho_{\Theta_1(\eta)}(\mathfrak{g}) \rangle, \langle \Delta_{\Theta_1(\eta)}(\mathfrak{g}), \nabla_{\Theta_1(\eta)}(\mathfrak{g}) \rangle\} & \text{if } \eta \in \aleph_1 - \aleph_2 \\ \{\langle \Omega_{\Theta_2(\eta)}(\mathfrak{g}), \mho_{\Theta_2(\eta)}(\mathfrak{g}) \rangle, \langle \Delta_{\Theta_2(\eta)}(\mathfrak{g}), \nabla_{\Theta_2(\eta)}(\mathfrak{g}) \rangle\} & \text{if } \eta \in \aleph_2 - \aleph_1 \\ \{\langle Max\{\Omega_{\Theta_1(\eta)}(\mathfrak{g}), \Omega_{\Theta_2(\eta)}(\mathfrak{g})\}, Min\{\mho_{\Theta_1(\eta)}(\mathfrak{g}), \mho_{\Theta_2(\eta)}(\mathfrak{g})\} \rangle, & \text{if } \eta \in \aleph_1 \cap \aleph_2 \\ \langle Max\{\Delta_{\Theta_1(\eta)}(\mathfrak{g}), \Delta_{\Theta_2(\eta)}(\mathfrak{g})\}, Min\{\nabla_{\Theta_1(\eta)}(\mathfrak{g}), \nabla_{\Theta_2(\eta)}(\mathfrak{g})\} \rangle\} \end{cases}$$

suppose,

$(\Theta_1, \aleph_1) \subseteq (\Theta_2, \aleph_2)$. Then $\aleph_1 \subseteq \aleph_2$ and

$\Omega_{\Theta_1(\eta)}(\mathfrak{g}) \leq \Omega_{\Theta_2(\eta)}(\mathfrak{g}), \mho_{\Theta_2(\eta)}(\mathfrak{g}) \leq \mho_{\Theta_1(\eta)}(\mathfrak{g}),$

$\Delta_{\Theta_1(\eta)}(\mathfrak{g}) \leq \Delta_{\Theta_2(\eta)}(\mathfrak{g}), \nabla_{\Theta_2(\eta)}(\mathfrak{g}) \leq \nabla_{\Theta_1(\eta)}(\mathfrak{g})$, for every $\eta \in \aleph_1$ and $\mathfrak{g} \in \mathfrak{G}$

since $\aleph_1, \aleph_2 \subseteq \mathscr{E}_1 \times \mathscr{E}_2 \times \ldots \times \mathscr{E}_n$

Therefore for any $\eta_c \leq_{\aleph_1} \eta_d$ we have $\Theta_1(\eta_c) \subseteq \Theta_1(\eta_d), \forall \eta_c, \eta_d \in \aleph_1$

and for any $\varsigma_c \leq_{\aleph_2} \varsigma_d$ we have $\Theta_2(\varsigma_c) \subseteq \Theta_2(\varsigma_d), \forall \varsigma_c, \varsigma_d \in \aleph_2$

Now for any $\varpi_c, \varpi_d \in \aleph_3$ and $\varpi_c \leq_{\aleph_3} \varpi_d$

$\Rightarrow \varpi_c, \varpi_d \in \aleph_1 \cup \aleph_2$

$\Rightarrow \varpi_c, \varpi_d \in \aleph_1 \cap \aleph_2$ or $\varpi_c, \varpi_d \in \aleph_2$ and $\varpi_c, \varpi_d \notin \aleph_1$ because $\aleph_1 \subseteq \aleph_2$

now take $\varpi_c, \varpi_d \in \aleph_1 \cap \aleph_2$

$\Rightarrow \varpi_c, \varpi_d \in \aleph_1$ and $\varpi_c, \varpi_d \in \aleph_2$

$\Rightarrow \Theta_1(\varpi_c) \subseteq \Theta_1(\varpi_d)$ and $\Theta_2(\varpi_c) \subseteq \Theta_2(\varpi_d)$ whenever $\varpi_c \leq_{\aleph_1} \varpi_d, \varpi_c \leq_{\aleph_2} \varpi_d$

$\Rightarrow \Omega_{\Theta_1(\varpi_c)}(\mathfrak{g}) \leq \Omega_{\Theta_1(\varpi_d)}(\mathfrak{g}), \Omega_{\Theta_2(\varpi_c)}(\mathfrak{g}) \leq \Omega_{\Theta_2(\varpi_d)}(\mathfrak{g})$

$\mho_{\Theta_1(\varpi_d)}(\mathfrak{g}) \leq \mho_{\Theta_1(\varpi_c)}(\mathfrak{g}), \mho_{\Theta_2(\varpi_d)}(\mathfrak{g}) \leq \mho_{\Theta_2(\varpi_c)}(\mathfrak{g})$

$\Delta_{\Theta_1(\varpi_c)}(\mathfrak{g}) \leq \Delta_{\Theta_1(\varpi_d)}(\mathfrak{g}), \Delta_{\Theta_2(\varpi_c)}(\mathfrak{g}) \leq \Delta_{\Theta_2(\varpi_d)}(\mathfrak{g})$

$\nabla_{\Theta_1(\varpi_d)}(\mathfrak{g}) \leq \nabla_{\Theta_1(\varpi_c)}(\mathfrak{g}), \nabla_{\Theta_2(\varpi_d)}(\mathfrak{g}) \leq \nabla_{\Theta_2(\varpi_c)}(\mathfrak{g})$

$\Rightarrow \mathrm{Max}\{\Omega_{\Theta_1(\varpi_c)}(\mathfrak{g}), \Omega_{\Theta_2(\varpi_c)}(\mathfrak{g})\} \leq \mathrm{Max}\{\Omega_{\Theta_1(\varpi_d)}(\mathfrak{g}), \Omega_{\Theta_2(\varpi_d)}(\mathfrak{g})\}$

$\mathrm{Min}\{\mho_{\Theta_1(\varpi_d)}(\mathfrak{g}), \mho_{\Theta_2(\varpi_d)}(\mathfrak{g})\} \leq \mathrm{Min}\{\mho_{\Theta_1(\varpi_c)}(\mathfrak{g}), \mho_{\Theta_2(\varpi_c)}(\mathfrak{g})\}$

$$\mathrm{Max}\{\Delta_{\Theta_1(\varpi_c)}(\mathfrak{g}),\Delta_{\Theta_2(\varpi_c)}(\mathfrak{g})\} \le \mathrm{Max}\{\Delta_{\Theta_1(\varpi_d)}(\mathfrak{g}),\Delta_{\Theta_2(\varpi_d)}(\mathfrak{g})\}$$
$$\mathrm{Min}\{\nabla_{\Theta_1(\varpi_d)}(\mathfrak{g}),\nabla_{\Theta_2(\varpi_d)}(\mathfrak{g})\} \le \mathrm{Min}\{\nabla_{\Theta_1(\varpi_c)}(\mathfrak{g}),\nabla_{\Theta_2(\varpi_c)}(\mathfrak{g})\}$$
$$\Rightarrow \Omega_{\Theta_1(\varpi_c)\cup\Theta_2(\varpi_c)}(\mathfrak{g}) \le \Omega_{\Theta_1(\varpi_d)\cup\Theta_2(\varpi_d)}(\mathfrak{g})$$
$$\mho_{\Theta_1(\varpi_d)\cup\Theta_2(\varpi_d)}(\mathfrak{g}) \le \mho_{\Theta_1(\varpi_c)\cup\Theta_2(\varpi_c)}(\mathfrak{g})$$
$$\Delta_{\Theta_1(\varpi_c)\cup\Theta_2(\varpi_c)}(\mathfrak{g}) \le \Delta_{\Theta_1(\varpi_d)\cup\Theta_2(\varpi_d)}(\mathfrak{g})$$
$$\nabla_{\Theta_1(\varpi_d)\cup\Theta_2(\varpi_d)}(\mathfrak{g}) \le \nabla_{\Theta_1(\varpi_c)\cup\Theta_2(\varpi_c)}(\mathfrak{g})$$
$$\Rightarrow \Omega_{\Theta_3(\varpi_c)}(\mathfrak{g}) \le \Omega_{\Theta_3(\varpi_d)}(\mathfrak{g})$$
$$\mho_{\Theta_3(\varpi_d)}(\mathfrak{g}) \le \mho_{\Theta_3(\varpi_c)}(\mathfrak{g})$$
$$\Delta_{\Theta_3(\varpi_c)}(\mathfrak{g}) \le \Delta_{\Theta_3(\varpi_d)}(\mathfrak{g})$$
$$\nabla_{\Theta_3(\varpi_d)}(\mathfrak{g}) \le \nabla_{\Theta_3(\varpi_c)}(\mathfrak{g})$$
$$\Rightarrow \Theta_3(\varpi_c) \subseteq \Theta_3(\varpi_d) \text{ for } \varpi_c \le_{\aleph_3} \varpi_d$$

Thus $(\Theta_1, \aleph_1) \cup_{EXT} (\Theta_2, \aleph_2) \in$ LOq-RLDFHSS $(\mathfrak{G})$ if $\varpi_c, \varpi_d \in \aleph_1 \cap \aleph_2$

Now suppose for any $\varpi_c, \varpi_d \in \aleph_2, \varpi_c, \varpi_d \notin \aleph_1$ and $\varpi_c \le_{\aleph_2} \varpi_d$

$$\Rightarrow \Theta_2(\varpi_c) \subseteq \Theta_2(\varpi_d) \text{ whenever } \varpi_c \le_{\aleph_2} \varpi_d$$
$$\Rightarrow (\Theta_1, \aleph_1) \cup_{EXT} (\Theta_2, \aleph_2) \in \text{LOq-RLDFHSS } (\mathfrak{G})$$

Hence $(\Theta_1, \aleph_1) \cup_{EXT} (\Theta_2, \aleph_2) \in$ LOq-RLDFHSS $(\mathfrak{G})$ in both cases
$(\Theta_1, \aleph_1) \cup_{EXT} (\Theta_2, \aleph_2) \in$ LOq-RLDFHSS $(\mathfrak{G})$, if one of them is a LOq-RLDFHS subset of other. $\square$

**Proof of proposition 3.9**

*Proof.* Let $(\Theta_1, \aleph_1),(\Theta_2, \aleph_2) \in$ LOq-RLDFHSS $(\mathfrak{G})$. Then by Definition 3.9
$(\Theta_1, \aleph_1) \wedge (\Theta_2, \aleph_2) = (\Xi, \aleph_1 \times \aleph_2)$ also
$$\Xi(\eta,\varsigma)(\mathfrak{g}) = \{\langle Min\{\Omega_{\Theta_1(\eta)}(\mathfrak{g}),\Omega_{\Theta_2(\varsigma)}(\mathfrak{g})\}, Max\{\mho_{\Theta_1(\eta)}(\mathfrak{g}),\mho_{\Theta_2(\varsigma)}(\mathfrak{g})\}\rangle,$$
$$\langle Min\{\Delta_{\Theta_1(\eta)}(\mathfrak{g}),\Delta_{\Theta_2(\varsigma)}(\mathfrak{g})\}, Max\{\nabla_{\Theta_1(\eta)}(\mathfrak{g}),\nabla_{\Theta_2(\varsigma)}(\mathfrak{g})\}\rangle\}$$

For any $\eta_c \le_{\aleph_1} \eta_d$ we have $\Theta_1(\eta_c) \subseteq \Theta_1(\eta_d), \forall \eta_c, \eta_d \in \aleph_1$
and for any $\varsigma_c \le_{\aleph_2} \varsigma_d$ we have $\Theta_2(\varsigma_c) \subseteq \Theta_2(\varsigma_d), \forall \varsigma_c, \varsigma_d \in \aleph_2$
Now for any $(\eta_c, \varsigma_c),(\eta_d, \varsigma_d) \in \aleph_1 \times \aleph_2$. Then by Definition 3.8
The order on $\aleph_1 \times \aleph_2$ is $(\eta_c, \varsigma_c) \le_{\aleph_1 \times \aleph_2} (\eta_d, \varsigma_d) \Leftrightarrow \eta_c \le_{\aleph_1} \eta_d$ and $\varsigma_c \le_{\aleph_1} \varsigma_d$

$$\Rightarrow \Theta_1(\eta_c) \subseteq \Theta_1(\eta_d) \text{ and } \Theta_2(\varsigma_c) \subseteq \Theta_2(\varsigma_d)$$
$$\Rightarrow \Omega_{\Theta_1(\eta_c)}(\mathfrak{g}) \le \Omega_{\Theta_1(\eta_d)}(\mathfrak{g}), \Omega_{\Theta_2(\varsigma_c)}(\mathfrak{g}) \le \Omega_{\Theta_2(\varsigma_d)}(\mathfrak{g})$$
$$\mho_{\Theta_1(\eta_d)}(\mathfrak{g}) \le \mho_{\Theta_1(\eta_c)}(\mathfrak{g}), \mho_{\Theta_2(\varsigma_d)}(\mathfrak{g}) \le \mho_{\Theta_2(\varsigma_c)}(\mathfrak{g})$$
$$\Delta_{\Theta_1(\eta_c)}(\mathfrak{g}) \le \Delta_{\Theta_1(\eta_d)}(\mathfrak{g}), \Delta_{\Theta_2(\varsigma_c)}(\mathfrak{g}) \le \Delta_{\Theta_2(\varsigma_d)}(\mathfrak{g})$$
$$\nabla_{\Theta_1(\eta_d)}(\mathfrak{g}) \le \nabla_{\Theta_1(\eta_c)}(\mathfrak{g}), \nabla_{\Theta_2(\varsigma_d)}(\mathfrak{g}) \le \nabla_{\Theta_2(\varsigma_c)}(\mathfrak{g})$$
$$\Rightarrow \mathrm{Min}\{\Omega_{\Theta_1(\eta_c)}(\mathfrak{g}),\Omega_{\Theta_2(\varsigma_c)}(\mathfrak{g})\} \le \mathrm{Min}\{\Omega_{\Theta_1(\eta_d)}(\mathfrak{g}),\Omega_{\Theta_2(\varsigma_d)}(\mathfrak{g})\}$$
$$\mathrm{Max}\{\mho_{\Theta_1(\eta_d)}(\mathfrak{g}),\mho_{\Theta_2(\varsigma_d)}(\mathfrak{g})\} \le \mathrm{Max}\{\mho_{\Theta_1(\eta_c)}(\mathfrak{g}),\mho_{\Theta_2(\varsigma_c)}(\mathfrak{g})\}$$
$$\mathrm{Min}\{\Delta_{\Theta_1(\eta_c)}(\mathfrak{g}),\Delta_{\Theta_2(\varsigma_c)}(\mathfrak{g})\} \le \mathrm{Min}\{\Delta_{\Theta_1(\eta_d)}(\mathfrak{g}),\Delta_{\Theta_2(\varsigma_d)}(\mathfrak{g})\}$$
$$\mathrm{Max}\{\nabla_{\Theta_1(\eta_d)}(\mathfrak{g}),\nabla_{\Theta_2(\varsigma_d)}(\mathfrak{g})\} \le \mathrm{Max}\{\nabla_{\Theta_1(\eta_c)}(\mathfrak{g}),\nabla_{\Theta_2(\varsigma_c)}(\mathfrak{g})\}$$
$$\Rightarrow \Omega_{\Xi(\eta_c,\varsigma_c)}(\mathfrak{g}) \le \Omega_{\Xi(\eta_d,\varsigma_d)}(\mathfrak{g})$$
$$\mho_{\Xi(\eta_d,\varsigma_d)}(\mathfrak{g}) \le \mho_{\Xi(\eta_c,\varsigma_c)}(\mathfrak{g})$$
$$\Delta_{\Xi(\eta_c,\varsigma_c)}(\mathfrak{g}) \le \Delta_{\Xi(\eta_d,\varsigma_d)}(\mathfrak{g})$$
$$\nabla_{\Xi(\eta_d,\varsigma_d)}(\mathfrak{g}) \le \nabla_{\Xi(\eta_c,\varsigma_c)}(\mathfrak{g})$$

$$\Rightarrow \Xi(\eta_c, \varsigma_c) \subseteq \Xi(\eta_d, \varsigma_d) \text{ for } (\eta_c, \varsigma_c) \leq_{\aleph_1 \times \aleph_2} (\eta_d, \varsigma_d)$$

Therefore, $(\Theta_1, \aleph_1) \wedge (\Theta_2, \aleph_2) \in$ LOq-RLDFHSS $(\mathfrak{G})$. $\square$

**Proof of proposition 3.11**

*Proof.* Let $(\Theta_1, \aleph_1), (\Theta_2, \aleph_2) \in$ LOq-RLDFHSS $(\mathfrak{G})$. Then by Definition 3.11

$(\Theta_1, \aleph_1) \vee (\Theta_2, \aleph_2) = (\Xi, \aleph_1 \times \aleph_2)$ also

$$\Xi(\eta, \varsigma)(\mathfrak{g}) = \{\langle Max\{\Omega_{\Theta_1(\eta)}(\mathfrak{g}), \Omega_{\Theta_2(\varsigma)}(\mathfrak{g})\}, Min\{\mho_{\Theta_1(\eta)}(\mathfrak{g}), \mho_{\Theta_2(\varsigma)}(\mathfrak{g})\}\rangle,$$
$$\langle Max\{\Delta_{\Theta_1(\eta)}(\mathfrak{g}), \Delta_{\Theta_2(\varsigma)}(\mathfrak{g})\}, Min\{\nabla_{\Theta_1(\eta)}(\mathfrak{g}), \nabla_{\Theta_2(\varsigma)}(\mathfrak{g})\}\rangle\}$$

For any $\eta_c \leq_{\aleph_1} \eta_d$ we have $\Theta_1(\eta_c) \subseteq \Theta_1(\eta_d), \forall \eta_c, \eta_d \in \aleph_1$

and for any $\varsigma_c \leq_{\aleph_2} \varsigma_d$ we have $\Theta_2(\varsigma_c) \subseteq \Theta_2(\varsigma_d), \forall \varsigma_c, \varsigma_d \in \aleph_2$

Now for any $(\eta_c, \varsigma_c), (\eta_d, \varsigma_d) \in \aleph_1 \times \aleph_2$. Then by Definition 3.8

Now for any $(\eta_c, \varsigma_c), (\eta_d, \varsigma_d) \in \aleph_1 \times \aleph_2$. Then by Definition 3.8

The order on $\aleph_1 \times \aleph_2$ is $(\eta_c, \varsigma_c) \leq_{\aleph_1 \times \aleph_2} (\eta_d, \varsigma_d) \Leftrightarrow \eta_c \leq_{\aleph_1} \eta_d$ and $\varsigma_c \leq_{\aleph_1} \varsigma_d$

$$\Rightarrow \Theta_1(\eta_c) \subseteq \Theta_1(\eta_d) \text{ and } \Theta_2(\varsigma_c) \subseteq \Theta_2(\varsigma_d)$$

$$\Rightarrow \Omega_{\Theta_1(\eta_c)}(\mathfrak{g}) \leq \Omega_{\Theta_1(\eta_d)}(\mathfrak{g}), \Omega_{\Theta_2(\varsigma_c)}(\mathfrak{g}) \leq \Omega_{\Theta_2(\varsigma_d)}(\mathfrak{g})$$
$$\mho_{\Theta_1(\eta_d)}(\mathfrak{g}) \leq \mho_{\Theta_1(\eta_c)}(\mathfrak{g}), \mho_{\Theta_2(\varsigma_d)}(\mathfrak{g}) \leq \mho_{\Theta_2(\varsigma_c)}(\mathfrak{g})$$
$$\Delta_{\Theta_1(\eta_c)}(\mathfrak{g}) \leq \Delta_{\Theta_1(\eta_d)}(\mathfrak{g}), \Delta_{\Theta_2(\varsigma_c)}(\mathfrak{g}) \leq \Delta_{\Theta_2(\varsigma_d)}(\mathfrak{g})$$
$$\nabla_{\Theta_1(\eta_d)}(\mathfrak{g}) \leq \nabla_{\Theta_1(\eta_c)}(\mathfrak{g}), \nabla_{\Theta_2(\varsigma_d)}(\mathfrak{g}) \leq \nabla_{\Theta_2(\varsigma_c)}(\mathfrak{g})$$

$$\Rightarrow Max\{\Omega_{\Theta_1(\eta_c)}(\mathfrak{g}), \Omega_{\Theta_2(\varsigma_c)}(\mathfrak{g})\} \leq Max\{\Omega_{\Theta_1(\eta_d)}(\mathfrak{g}), \Omega_{\Theta_2(\varsigma_d)}(\mathfrak{g})\}$$
$$Min\{\mho_{\Theta_1(\eta_d)}(\mathfrak{g}), \mho_{\Theta_2(\varsigma_d)}(\mathfrak{g})\} \leq Min\{\mho_{\Theta_1(\eta_c)}(\mathfrak{g}), \mho_{\Theta_2(\varsigma_c)}(\mathfrak{g})\}$$
$$Max\{\Delta_{\Theta_1(\eta_c)}(\mathfrak{g}), \Delta_{\Theta_2(\varsigma_c)}(\mathfrak{g})\} \leq Max\{\Delta_{\Theta_1(\eta_d)}(\mathfrak{g}), \Delta_{\Theta_2(\varsigma_d)}(\mathfrak{g})\}$$
$$Min\{\nabla_{\Theta_1(\eta_d)}(\mathfrak{g}), \nabla_{\Theta_2(\varsigma_d)}(\mathfrak{g})\} \leq Min\{\nabla_{\Theta_1(\eta_c)}(\mathfrak{g}), \nabla_{\Theta_2(\varsigma_c)}(\mathfrak{g})\}$$

$$\Rightarrow \Omega_{\Xi(\eta_c, \varsigma_c)}(\mathfrak{g}) \leq \Omega_{\Xi(\eta_d, \varsigma_d)}(\mathfrak{g})$$
$$\mho_{\Xi(\eta_d, \varsigma_d)}(\mathfrak{g}) \leq \mho_{\Xi(\eta_c, \varsigma_c)}(\mathfrak{g})$$
$$\Delta_{\Xi(\eta_c, \varsigma_c)}(\mathfrak{g}) \leq \Delta_{\Xi(\eta_d, \varsigma_d)}(\mathfrak{g})$$
$$\nabla_{\Xi(\eta_d, \varsigma_d)}(\mathfrak{g}) \leq \nabla_{\Xi(\eta_c, \varsigma_c)}(\mathfrak{g})$$

$$\Rightarrow \Xi(\eta_c, \varsigma_c) \subseteq \Xi(\eta_d, \varsigma_d) \text{ for } (\eta_c, \varsigma_c) \leq_{\aleph_1 \times \aleph_2} (\eta_d, \varsigma_d)$$

Therefore, $(\Theta_1, \aleph_1) \vee (\Theta_2, \aleph_2) \in$ LOq-RLDFHSS $(\mathfrak{G})$. $\square$

### Funding

This work was supported by the National Research Foundation of Korea (NRF) grant funded by the Korea government (MSIT) (No. RS-2023-00277907) and by the MSIT (Ministry of Science and ICT), Korea, under the ITRC (Information Technology Research Center) support program (IITP-2024-RS-2024-00438335) supervised by the IITP (Institute for Information & Communications Technology Planning & Evaluation). There was no additional external funding received for this study. The funders had no role in study design, data collection and analysis, decision to publish, or preparation of the manuscript.

## Grant Disclosures
The following grant information was disclosed by the authors:
National Research Foundation of Korea (NRF)—Korea government (MSIT): RS-2023-00277907.
MSIT (Ministry of Science and ICT), Korea, under the ITRC (Information Technology Research Center): IITP-2024-RS-2024-00438335.
IITP (Institute for Information & Communications Technology Planning & Evaluation).

## Competing Interests
Dragan Pamucar is an Academic Editor for PeerJ.

## Author Contributions
- J. Vimala conceived and designed the experiments, performed the experiments, analyzed the data, prepared figures and/or tables, and approved the final draft.
- A. N. Surya conceived and designed the experiments, analyzed the data, authored or reviewed drafts of the article, and approved the final draft.
- Nasreen Kausar performed the experiments, authored or reviewed drafts of the article, and approved the final draft.
- Dragan Pamucar analyzed the data, authored or reviewed drafts of the article, and approved the final draft.
- Seifedine Kadry performed the computation work, prepared figures and/or tables, and approved the final draft.
- Jungeun Kim performed the experiments, performed the computation work, prepared figures and/or tables, and approved the final draft.

## Data Availability
The pseudocode is available in Figs. 1–3.

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
