# Peer review of "Hybrid decision support system disaster management: application of lattice ordered q-rung linear Diophantine fuzzy hypersoft sets"

_PeerJ Computer Science, doi:10.7717/peerj-cs.2927_

## Round 0.1 · original submission · Major Revisions

You must address all comments of the 3 reviewers

Reviewer 1 ·

Basic reporting

The manuscript titled "Hybrid Decision Support System Disaster Management: Application of Lattice Ordered q-Rung Linear Diophantine Fuzzy Hypersoft Sets" is well written, well presented, and is mathematically correct.

Experimental design

Research question well defined, relevant & meaningful. It is stated how research fills an identified knowledge gap

Validity of the findings

All underlying data have been provided; they are robust, statistically sound, & controlled.

Additional comments

(1). The authors use many abbreviations. I would suggest creating an appendix with their list so it is easier for the reader to find their meaning.
(2). This manuscript needs further improvement in sentence expression because of a few grammar mistakes.
(3). Please add some more future possible extensions of this research work.
(4). Give equation numbering to important results.
(5). Please add a state-of-the-art literature review of linear Diophantine fuzzy sets and q-rung orthopair fuzzy sets

Reviewer 2 ·

Basic reporting

1. There are spelling/ grammatical mistakes in the article some of them are given below:
*. "a greater number" instead of "more number" (line 31)
*. "decision-making(MADM)" must be "decision-making (MADM)" (line 38)
*. The spellings of characteristics are wrong (line 186, line 341)
*. The sentence is not making sense (Line 298-299)
*. In (line 303)"National Response Framework" website link in footnote
*. In (Line 312) please add a concise description of each cited research highlighting particular areas/methodology in decision making.
*. In (Line 319) The starting sentence is grammatically incorrect.
*. In (Line 355) Step 2, and the caption of Table 2 the spelling of "comparison" is wrong
2. The caption of "Figure 2" is very short and brief please provide a detailed caption for the comprehension of the reader e.g. "Flowchart showing the steps of the proposed LOq-RLDFHSS-based MADM algorithm". Similarly, the captions of Table 1, Table 2, Table 3, and Table 4 are not elaborative and provide detailed captions for each table.

Experimental design

The literature review seems appropriate and well structured, but it is missing the review of the recent idea of "quadratic Diophantine fuzzy sets" https://doi.org/10.3934/math.2023738 , please add it to your literature review.
In example 1:
1. (Line 172 and 173), It is unclear how the authors define the relation between the elements (Linguistic terms), is it an assumed relation (provide reason for assuming) or is there some formulation behind such relation (provide the formula)? same thing happened in the "4.2.2 problem" (Lines 326 and 327)
2. (Line 182) The authors considered only two combinations of the cartesian product and neglected the possibilities for example: (cheap, good, dissatisfies) or (not good, cheap, dissatisfied), etc. Is there any particular reason for that?
3. In this example and in "4.2.2 problem" the authors considered q=3, is there any particular reason for that?
4. In (line 188) it seems that the assertion theta(eta2) doesn't contain in theta(eta4). kindly remove this error (interchange 0.6 with 0.7 under the g2 non-membership function).

In section 3, an example at the end of the section is necessary to elaborate the definitions and results presented within the section.

Validity of the findings

In section 4
1. Definition 4.1 Can the authors, define the range (in interval form) of hac (entries of comparison matrix) (An idea is to compute min max of hac)? If so, then in definition 4.2 the range for the score function could also be defined.
2. In (line 347) how could percentages 33+87=120 exceed 100 please provide concrete justification for that.
3. Step 5 of the algorithm shows a limitation of the algorithm in the case of ties (the algorithm is incapable of breaking ties). To overcome this use the hesitancy function (see the idea of hesitancy function for linear Diophantine fuzzy sets in " https://doi.org/10.1016/j.engappai.2024.107953 " to refer to it).
In section 5
1. In line 367 the term referring to "transient approach" needs clarification/elaboration.
2. In (lines 371-374) No numerical details of the worse alternative g-hat are provided, please provide the details of g-hat.
3. In Table 4 first column describes methods its title should be "DM methods (based on)" and it entries must be FS [reference], IFS [reference], PFS [reference], etc. (to remove unnecessary repetition) also add references inside the table and remove them from the subsection "Superiority of the proposed MADM method".
4. Expand the analysis to discuss the practical implications of the proposed method in terms of computational efficiency and scalability.

In section 6
1. The conclusion consists of overly generic concluding statements and Specific future directions regarding societal problems are missing, adding them will make the conclusion more impactful

Additional comments

The article demonstrates a significant contribution to the field of decision-making under uncertainty.
I recommend minor revisions for language clarity and the inclusion of a brief discussion on ethical considerations, and major revisions for the examples, and results clarities specifically for sections 4 and 5.

Reviewer 3 ·

Basic reporting

Review report
Hybrid decision support system disaster management: Application of lattice ordered Q-rung linear Diophantine fuzzy hypersoft sets
I think it is good work for the journal but some suggestions are listed as below before going to the next round.

1. The abstract only contains some sentences without any process conditions, which is insufficient to delineate the whole picture of contribution of this study.

2. Introduction" sections of the manuscript are not well organized. This section can be made much more impressive by highlighting the contributions.

3. There are some grammatical errors. Please check the whole manuscript to improve the language.

4. Give a clear motivation of the paper in introduction section.

5. In introduction, the authors should add references to articles of fuzzy sets.

6. Throughout my reading, I met some typos. The authors are suggested to check them carefully to improve the quality of the manuscript.

7. I suggest the authors make their paper simple in the content, so as to let readers understand the main contribution easily.

8. Add future work direction in conclusion section.

Experimental design

No comments

Validity of the findings

No Comments

Additional comments

No Comments

---

## Round 0.2 · Minor Revisions

As you can see, some minor comments must still be addressed

Reviewer 1 ·

Basic reporting

The authors revised the manuscript in a good manner. The manuscript should be accepted in its current firm.

Experimental design

The authors revised the manuscript in a good manner. The manuscript should be accepted in its current firm.

Validity of the findings

The authors revised the manuscript in a good manner. The manuscript should be accepted in its current firm.

Additional comments

nill

Reviewer 2 ·

Basic reporting

Concerns are addressed successfully.

Experimental design

Concerns are addressed successfully.

Validity of the findings

Concerns are addressed successfully, but in Table 4, the first column may be:
* * *
DM methods
(based
on)
* * *
FS
Zadeh (1965)
* * *
IFS
Atanassov (1986)
.
.
.
so on.

Don't repeat "DM methods based on" in every entry.

Additional comments

The article may be accepted, after the mentioned minor revision.

---

## Round 0.3 · Minor Revisions

Two reviewers have recommended acceptance. However, I have some more comments of my own. I was wondering whether I may kindly ask you to prepare the final revision. Please see the attached file.

**Language Note:** The Academic Editor has identified that the English language must be improved. PeerJ can provide language editing services - please contact us at [email protected] for pricing (be sure to provide your manuscript number and title). Alternatively, you should make your own arrangements to improve the language quality and provide details in your response letter. – PeerJ Staff

Reviewer 2 ·

Basic reporting

All concerns are addressed successfully, the manuscript can be accepted.

Experimental design

All concerns are addressed successfully, the manuscript can be accepted.

Validity of the findings

All concerns are addressed successfully, the manuscript can be accepted.

Reviewer 3 ·

Basic reporting

The questions are resolved, and it can be accepted.

Experimental design

No comments

Validity of the findings

No comment

---

## Round 0.4 · Minor Revisions

Response 1: Use the list below to review your submission and identify any errors that an automatic tool has flagged. This automated tool was used only until page 10, and I expect you to review the entire submission carefully.

Responses 2 to 7: I thank you for revising the submission.
* * *
p.1, l.26: "The discovery of lattice-ordered q-rung linear Diophantine fuzzy hypersoft set is a significant extension" should be "The discovery of the lattice-ordered q-rung linear Diophantine fuzzy hypersoft set is a significant extension" (missing article "the").

p.1, l.27: "Further, an algorithm based on proposed operations is presented" should be "Further, an algorithm based on the proposed operations is presented" (missing article "the").

p.1, l.28: "helps in choosing the most appropriate plan to tackle the known natural disaster by considering greater number of attributes together" should be "helps in choosing the most appropriate plan to tackle the known natural disaster by considering a greater number of attributes together" (missing article "a").

p.2, l.43: "set(FS) theory introduced by ZadehZadeh (1965) in 1965" should be "set (FS) theory introduced by Zadeh (1965)" (repetition of author name and year, missing space after "set").

p.2, l.44: "To overcome these restrictions, AtanassovAtanassov (1986) devised the notion" should be "To overcome these restrictions, Atanassov (1986) devised the notion" (duplicate author name).

p.2, l.47: "YagerYager (2013) developed the Pythagorean fuzzy set" should be "Yager (2013) developed the Pythagorean fuzzy set" (duplicate author name).

p.2, l.48: "YagerYager (2016) also proposed" should be "Yager (2016) also proposed" (duplicate author name).

p.2, l.51: "Riaz and HashmiRiaz and Hashmi (2019)" should be "Riaz and Hashmi (2019)" (duplicate author names).

p.2, l.55: "Zia et al.Zia et al. (2023)" should be "Zia et al. (2023)" (duplicate citation).

p.2, l.56: "AlmagrabiAlmagrabi et al. (2022)" should be "Almagrabi et al. (2022)" (duplicate citation).

p.2, l.61: "MolodtsovMolodtsov (1999)" should be "Molodtsov (1999)" (duplicate citation).

p.2, l.62: "Maji et al.Roy and Maji (2007)" should be "Roy and Maji (2007)" (incorrect or mixed citation format).

p.2, l.65: "Agman and KaratasÇağman and Karataş (2013)" should be "Çağman and Karataş (2013)" (incorrect citation format).

p.2, l.66: "Peng et al.Peng et al. (2015)" should be "Peng et al. (2015)" (duplicate citation).

p.2, l.66: "Hussain et al.Hussain et al. (2020)" should be "Hussain et al. (2020)" (duplicate citation).

p.2, l.66: "Riaz et al.Riaz et al. (2020)" should be "Riaz et al. (2020)" (duplicate citation).

p.2, l.69: "SanrandacheSmarandache (2018)" should be "Smarandache (2018)" (misspelling and duplicate).

p.2, l.71: "SanrandacheSmarandache (2018)" should be "Smarandache (2018)" (duplicate author name again).

p.2, l.73: "S Khan et al.Khan et al. (2022)" should be "Khan et al. (2022)" (duplicate citation with inconsistent author naming).

p.2, l.74: "Surya et al.Surya et al. (2024)" should be "Surya et al. (2024)" (duplicate citation).

p.2, l.76: "Ali et al.Ali et al. (2015)" should be "Ali et al. (2015)" (duplicate citation).

p.2, l.77: "Aslam et al.Aslam et al. (2019)" should be "Aslam et al. (2019)" (duplicate citation).

p.2, l.78: "Mahmood et al.Mahmood et al. (2018a)" should be "Mahmood et al. (2018a)" (duplicate citation).

p.3, l.96: "by the proposed MADM approach since the existing DM methods in the disaster management field cannot handle multiple attributes together" should be "by the proposed MADM approach, since existing DM methods in the disaster management field cannot effectively handle multiple attributes simultaneously" (missing comma; awkward phrasing).

p.3, l.108: "Based on the LOq-RLDFHSS, a MADM algorithm is presented in this study" would be clearer as "A MADM algorithm based on the LOq-RLDFHSS is presented in this study" (sentence reordering for clarity).

p.3, l.116: "The list of most of the abbreviations used in this study is given as a Table in Appendix A" should be "The list of most abbreviations used in this study is provided in Table A in the Appendix" (awkward phrasing and article usage).

p.3, l.117: "The paper is Structured as follows" should be "The paper is structured as follows" (unnecessary capitalization of "Structured").

p.3, l.120: "shows the efficiency of the proposed algorithm" should be "which demonstrates the efficiency of the proposed algorithm" (add relative clause for better sentence connection).

p.4, l.126: "A binary relation ξ on a non-empty set A is said to be partial order" should be "A binary relation ξ on a non-empty set A is said to be a partial order" (missing article "a").

p.4, l.127: "Also, ξ is said to be total order on A" should be "Also, ξ is said to be a total order on A" (missing article "a").

p.4, l.128: "A partial order set L is said to be a lattice" should be "A partially ordered set L is said to be a lattice" (incorrect terminology; "partially ordered" is the correct expression).

p.4, l.138: "Then (Q; A) is said to be LOSS" should be "Then (Q; A) is said to be a LOSS" (missing article "a").

p.4, l.143: "Then (Q; A) is said to be LOIFSS" should be "Then (Q; A) is said to be a LOIFSS" (missing article "a").

p.4, l.148: "It can be written as (Q; A₁) = { (h; Q(h)) : h ∈ A₁; Q(h) ∈ P(G) }" should be introduced with a more precise sentence like "This can be represented as..." for smoother flow.

p.4, l.154: "Then q-Rung Linear Diophantine Fuzzy Hypersoft Set over G (q-RLDFHSS(G)) is the pair (Q; A₁) defined by the map" should be "Then, the q-Rung Linear Diophantine Fuzzy Hypersoft Set over G (q-RLDFHSS(G)) is the pair (Q; A₁), defined by the map..." (missing article, optional comma for flow).

p.5, l.175: "Moreover, the example that follows will make it easier to understand" should be "The following example clarifies the definition above" (improve precision and tone for formal writing).

p.5, l.181: "The elements in each set A1; A2 and A3 has an order among them" should be "The elements in each of the sets A1, A2, and A3 have an order among them" (subject-verb agreement; missing article).

p.5, l.189–192: "The attribute 'charges' and its attribute values exemplify that the alternative is cheap or not cheap" could be rewritten as "The attribute 'charges' indicates whether the alternative is considered cheap or not" (the original sentence is repetitive and awkward).

p.5, l.193: "Then, the cartesian product of attribute values exemplifies that the alternative is (cheap, good, satisfies) altogether or (not cheap, not good, dissatisfies) altogether" should be "The Cartesian product of attribute values indicates whether the alternative jointly satisfies the conditions (cheap, good, satisfies) or their opposites" (awkward structure; improve clarity and flow).

p.6, l.199: "so (Q; A1) is a LOq-RLDFHSS (G)" should be "therefore, (Q; A1) is a LOq-RLDFHSS (G)" (improve logical transition).

p.6, l.220: "Then their Extended union is defined by" should be "Their extended union is defined as follows" (unnecessary "then"; normalize capitalization).

p.6, l.223: "Then (Q1; A1) ∪EXT (Q2; A2) ∈ LOq-RLDFHSS(G), if one of them is a LOq-RLDFHS subset of other" should be "Then (Q1; A1) ∪EXT (Q2; A2) ∈ LOq-RLDFHSS(G), if one is a LOq-RLDFHSS subset of the other" (missing article and incorrect abbreviation).

p.7, l.228: "Then their 'AND' operation is defined by" should be "Their 'AND' operation is defined as follows" (avoid using "then" and improve clarity).

p.7, l.247: "Then (W₁; A₁) is called relative null LOq-RLDFHSS and denoted by /0A₁" should be "Then, (W₁; A₁) is called the relative null LOq-RLDFHSS and is denoted by ∅A₁" (add article and standard symbol for null set).

p.7, l.250: "Then (Q₁; A₁) is called relative universal LOq-RLDFHSS and denoted by U_A1" should be "Then, (Q₁; A₁) is called the relative universal LOq-RLDFHSS and is denoted by UA₁" (add article "the" and improve phrasing for clarity).

p.7, l.260: "Then complement of (Q₁; A₁) denoted by (Q₁; A₁)^c and is defined by" should be "The complement of (Q₁; A₁), denoted by (Q₁; A₁)^c, is defined as follows" (correct syntax and improve flow).

p.8, l.277: Each item starting with "Ł" (bullet points) lacks proper introductory sentence or integration into the paragraph structure. A suggested rewrite: "The following operations are then derived:"

p.9, l.278: "In this section, the comparison matrix of LOq-RLDFHSS and a MADM algorithm based on LOq-RLDFHSS are described" should be "In this section, the comparison matrix of LOq-RLDFHSS and a MADM algorithm based on it are described" (remove redundancy and improve flow).

p.9, l.282: "rows are elements of alternatives such as g₁; g₂; …; gₘ and columns are the parameters h₁; h₂; …; hᵣ" should be "rows represent the alternatives g₁, g₂, …, gₘ, and columns represent the parameters h₁, h₂, …, hᵣ" (improve clarity and structure).

p.9, l.289: "range of h_ac lies within [-(m-1), m-1]" should be "the range of h_ac lies within [−(m−1), m−1]" (missing article and use of proper typographical symbols).

p.9, l.290: "Sₐ = ∑ h_ac Whose, the range lies within..." should be "Sₐ = ∑ h_ac, where the range lies within..." ("Whose" is incorrect in this context; use "where").

p.9, l.292: "The following is the algorithm for choosing the most suitable alternative" should be "The following steps describe the algorithm for selecting the most suitable alternative" (more formal and consistent wording).

p.9, l.297: "Choose any one if more than one maximum was obtained" should be "If multiple alternatives share the maximum score, select any one of them" (improve precision and grammar).

p.10, l.315: "Mitigation is 'sustained action that minimizes or prevents long-term danger to individuals and assets..." should be "Mitigation refers to sustained actions that minimize or prevent long-term danger to individuals and assets..." (correct article and improve grammatical structure).

p.10, l.324: "The recovery phase starts as soon as there is no further danger to human life" should be "The recovery phase begins once there is no immediate danger to human life" (refine formality and precision).

---

## Round 0.5 · Minor Revisions

Thank you for the revision. My AI-based tool still finds issues that it classifies as significant grammatical errors. Will you take a look?

1. p.1, l.43
Original: “The fuzzy set(FS) theory introduced by Zadeh (1965) in 1965 is very useful…”
Correction: Remove repetition → “introduced by Zadeh (1965)”

2. p.1, l.54
Original: “…as a resultMahmood et al. (2021a,b).”
Correction: Add space → “as a result Mahmood et al. (2021a,b).”

3. p.1, l.56
Original: “…set(q-RLDFS), a particular extension…”
Correction: Add space → “set (q-RLDFS), a particular extension…”

4. p.1, l.58
Original: “…real-world decision-making studies such company selection problemAli (2025)…”
Correction: Add “as” and space → “…such as company selection problem Ali (2025)…”

5. p.1, l.61
Original: “…soft set(SS) theory…”
Correction: Add space → “…soft set (SS) theory…”

6. p.1, l.62
Original: “…fuzzy soft set(FSS), which helps present fuzzy data…”
Correction: Add space → “…fuzzy soft set (FSS), which helps present…”

7. p.1, l.66
Original: “…intuitionistic fuzzy soft set(IFSS)…”
Correction: Add space → “…intuitionistic fuzzy soft set (IFSS)…”

8. p.1, l.67
Original: “…Pythagorean fuzzy soft set(PFSS)…”
Correction: Add space → “…Pythagorean fuzzy soft set (PFSS)…”

9. p.1, l.67
Original: “…q-rung orthopair fuzzy soft set(q-ROFSS)…”
Correction: Add space → “…q-rung orthopair fuzzy soft set (q-ROFSS)…”

10. p.1, l.67
Original: “…linear Diophantine fuzzy soft set(LDFSS).”
Correction: Add space → “…linear Diophantine fuzzy soft set (LDFSS).”

11. p.2, l.93
Original: “The study aims to close research gaps by developing many fundamental algebraic operations of LOq-RLDFHSS and a MADM method based on LOq-RLDFHSS.”
Correction: Reword for clarity → “The study aims to close these research gaps by developing fundamental algebraic operations and a MADM method based on LOq-RLDFHSS.”

12. p.10, l.306
Original: “A mix of efforts by individuals, households, businesses, local governments, and/or higher levels of government are typically required…”
Correction: Subject-verb agreement → “…a mix of efforts… is typically required…”

13. p.13, l.418
Original: “…the proposed MADM methodology is capable to handle problems…”
Correction: Improve phrasing → “…is capable of handling problems…”

14. p.13, l.419
Original: “…the results obtained by the proposed method is more relaiable and accurate…”
Correction: Fix typo → “…is more reliable and accurate…”

15. p.16, l.445
Original: “…it will be focused to develope formulation of advanced information measures…”
Correction: Fix verb form and typo → “…it will focus on developing advanced information measures…”

16. p.16, l.446
Original: “…it will be focused to overcome the limitations…”
Correction: Reword → “…it will focus on overcoming the limitations…”

17. p.16, l.447
Original: “…by uitilizing the concept of hesitancy function…”
Correction: Fix typo → “…by utilizing the concept of hesitancy function…”

---

## Round 0.6 · accepted · Accept

Thank you for fixing the typos.